rsos.royalsocietypublishing.org

organic chemistry/synthetic chemistry

trifluoromethanesulfonates, heteroatom reagents, annulation, cyclic compounds

**Author for correspondence:**
Chanjuan Xi
e-mail: cjxi@tsinghua.edu.cn

†These authors contributed equally to this study.

This article has been edited by the Royal Society of Chemistry, including the commissioning, peer review process and editorial aspects up to the point of acceptance.

# ROTf-induced annulation of heteroatom reagents and unsaturated substrates leading to cyclic compounds

Song Zou[1,†], Sheng Wang[1,†] and Chanjuan Xi[1,2]

[1]MOE Key Laboratory of Bioorganic Phosphorus Chemistry and Chemical Biology, Department of Chemistry, Tsinghua University, Beijing 100084, People's Republic of China
[2]State Key Laboratory of Elemento-Organic Chemistry, Nankai University, Tianjin 300071, People's Republic of China

CX, 0000-0002-9602-7309

The development of metal-free organic reactions is one of the hotspots in the synthesis of cyclic compounds. ROTf (alkyl trifluoromethanesulfonates), due to their good electrophilicity, are powerful alkylating reagents at heteroatoms such as nitrogen, oxygen, sulfur and phosphorus to induce an electrophilic centre for carbon–carbon or carbon–heteroatom bond formation. Inspired by this chemistry, a variety of research concentrating on heterocycles synthesis has been carried out. In this review, we mainly summarize the ROTf-induced annulation of heteroatom reagents such as nitriles, carbodiimides, azobenzenes, isothiocyanates, aldehydes, isocyanates and phosphaalkene with themselves or alkynes to afford cyclic compounds.

## 1. Introduction

Heterocycles are ubiquitous in natural products, pharmaceuticals, organic materials and numerous functional molecules. Therefore, organic chemists have been making extensive efforts to produce these heterocyclic compounds by developing new and efficient synthetic transformations. Transition-metal-catalysed reactions are some of the most attractive methodologies for synthesizing heterocycles, because a transition-metal-catalysed reaction can directly construct complicated molecules from readily accessible starting materials under mild conditions [1–6]. Nevertheless, transition-metal-catalysed reactions are still limited in applications and confront challenges to some extent, because transition-metals are expensive, toxic, inconvenient for operation and environment damage. In this regard, a transition-metal-free methodology for the construction of important heterocyclic compounds in drug discovery and material science has attracted attention [7–15]. ROTf

(*a*) ROTf-mediated substitution:

$$R'{-}X + \color{red}{ROTf} \longrightarrow [R'{-}X{-}\color{red}{R}]^+\ ^-OTf \xrightarrow{Nu^-} R{-}Nu$$
$$\color{red}{R}X + {}^-OTf$$

(*b*) ROTf-mediated cyclization:

X= N, S, O, P    Y= C, N, P

**Scheme 1.** ROTf-mediated reactions.

(alkyl trifluoromethanesulfonates) are powerful alkylating reagents, which are frequently used in alkylation of nucleophiles [16–18]. Benefiting from fluorine in $^-$OTf to stabilize the negative charge, $^-$OTf possesses excellent leaving ability, which makes the ROTf much more reactive as alkylation reagents than alkyl iodide and $(MeO)_2SO_2$. As useful and versatile precursors in a variety of organic transformations, many methods have been developed to prepare ROTf, such as the reaction of orthoformates and triflic anhydrides under solvent-free condition [19–26]. The intrinsic electrophilicity of ROTf is often used as alkylation reagents for diverse substrates containing heteroatoms such as P [27], O [28], N [29–31], S [32,33], Te [34], Ge [35], Bi [36], Si [37] and Se [38,39], which straightforwardly affords the corresponding alkylated products with aid of base or stable triflates irreversibly that even could be applied as ionic liquids [40,41]. Moreover, the resulting triflates could be further transformed into other products by substitution [30,42–44] (scheme 1*a*). On the other hand, unsaturated heteroatom-containing reagents, such as nitriles, aldehydes, isocyanates and isothiocyanates, could be captured by ROTf to generate electrophiles bearing carbon cations [29,31,45,46], which could be subsequently reacted with appropriate unsaturated substrates to produce cyclic compounds by tandem electrophilic reactions/cyclization (scheme 1*b*); although HOTf or $HNTf_2$ as a catalyst or promoter also demonstrated excellent electrophilic cyclization involving alkynes or alkenes. However, preparing highly functionalized heterocycles is still difficult. ROTf-induced cyclization featuring metal-free, easy to handle and good selectivity provided a feasible approach to diverse heterocyclic compounds. In this review, we mainly focus on the ROTf-induced annulation of unsaturated heteroatom reagents such as nitriles, carbodiimides, azobenzenes, isothiocyanates, aldehydes, isocyanates and phosphaalkene with themselves or alkynes to afford cyclic compounds.

# 2. ROTf-induced annulation of nitrogen-containing substrates with unsaturated compounds

## 2.1. ROTf-induced annulation of nitriles

Nitriles as unsaturated heteroatom reagents could react with ROTf to form *N*-alkylated nitriliums, which were well investigated by Booth *et al.* [31] in 1980. However, electrophilicity of *N*-alkylated nitriliums has rarely been used in further reactions until recently. In 2014, our group [47] reported MeOTf-induced carboannulation of arylnitriles and aromatic alkynes to construct indenones **1** (scheme 2). Triflate **3** was isolated when 5-decyne was used indicating **I-1** might be an intermediate in this reaction. A range of functionalized indenone derivatives was obtained. When *ortho*-substituted arylnitriles were used, indenone imine **I-2** would further cyclize with another molecule of nitriliums to give indeno[1,2-*c*]-isoquinolines **2** with the construction of one carbocycle and one heterocycle. Although transition-metal-catalysed annulation of benzimide or arylcarbonyl and arylnitrile with alkynes to the formation of indenones has been reported (for examples, see [48–55]), this reaction reveals a simple reaction process for the synthesis of indenones under metal-free conditions.

Arylnitriles and alkynes could be induced by MeOTf to afford indenones via intermediate **I-5**. We envisioned the utilization of alkylnitriles, which lack an aryl group for the ring closure, might lead to a different way for ring formation. To our delight, the reaction of alkylnitriles, alkynes and MeOTf indeed afforded tetrasubstituted NH-pyrroles **4** with high regioselectivity. The structure includes one carbon from ROTf to join the pyrroles [56] (scheme 3). It is noteworthy that the cyclized pyrrole captures

**Scheme 2.** MeOTf-induced cyclization of arylnitriles and alkynes.

**Scheme 3.** ROTf-induced cyclization of alkylnitriles and alkynes.

another nitrilium leading to substituted 2-acyl-NH-pyrroles after hydrolysis. When EtOTf was used instead of MeOTf, the product **5** was obtained. This reaction provides a practical and convenient method for the synthesis of multiply substituted 2-acylpyrroles from readily available starting materials in a one-pot reaction.

Furthermore, when 1,2-diphenylethyne and 1,2-di-*p*-tolylethyne were employed to react with MeOTf and alkylnitriles at 130°C, the isoquinolines **6** were obtained in good yields via intermediate **I-9**. The

**6a** 70%  **6b** 67%  **6c** 72%  **6d** 40%

**6e** 45%  **6f** 75%  **6g** 71%

**Scheme 4.** MeOTf-induced cyclization of alkylnitriles and diarylalkynes.

representative results are summarized in scheme 4. In the cases, the Friedel–Crafts reaction is favoured to give 6-membered products.

## 2.2. ROTf-induced annulation of carbodiimides

More recently, ROTf-induced electrophilic cyclization was extended to carbodiimides. Our group demonstrated an efficient ROTf-triggered intermolecular cyclization of carbodiimides to afford a range of 2-amino-4-imino-quinazolines **7**, 2-aminoquinazolinones **8** and 2,4-diaminoquinazolines **9**, respectively, which are important motifs in pharmaceuticals [57]. When *N,N'*-diarylcarbodiimides were employed, the reaction proceeded smoothly to afford the corresponding 2-amino-4-imino-quinazolines **7**, which could be hydrolysed to generate the corresponding 2-aminoquinazolinones **8**. The representative results are shown in scheme 5.

To further extend the substrate scope, a combination of two different carbodiimides has been achieved. A range of diarylcarbodiimides and dialkylcarbodiimides was investigated under the optimized reaction condition. The representative results are shown in scheme 6. A plausible mechanism was proposed. First, the carbodiimide is methylated by MeOTf and subsequently attacked by another molecule of carbodiimide to give intermediate **I-11**. Then, the intramolecular nucleophilic attack takes place to afford the four-membered intermediate **I-12**, which generates carbenium **I-13** after ring opening via C–N bond cleavage. Finally, intramolecular Friedel–Crafts annulation occurs to form the corresponding quinazolinone imine **7**, which could give 2,4-diamino-quinazoline **9** (R = alkyl group) and 2-amino-quinazolinone **8** (R = aryl group) after hydrolysis. This annulation reaction appears a general entry to the synthesis of 2-aminoquinazolinones and 2,4-diaminoquinazolines in a one-pot reaction under metal-free conditions.

## 2.3. ROTf-induced annulation of azobenzenes

Apart from nitriles and carbodiimides, azobenzenes are also significant nitrogen-containing compounds, which possess Lewis alkalinity. In 2014, we demonstrated MeOTf-induced cyclization of azobenzenes by N=N bond cleavage with aid of TCQ (tetrachloro-1,4-benzoquinone) as oxidant to afford *N*-arylbenzimidazoles **10** [58] (scheme 7). When unsymmetrical azobenzenes were used, cyclization tends to occur on the electron-rich anisolyl ring (**10e–10g**). EtOTf could facilitate N=N bond cleavage as well to generate 2-methylbenzimidazole **10h**. Although the reaction mechanism is not clear, a plausible mechanism is shown in scheme 7. The carbon atom from MeOTf inserts into the N=N bond and then cyclization to form *N*-arylbenzimidazole. This is the first example of N=N bond cleavage by a light main group element.

**Scheme 5.** ROTf-induced cyclization of diarylcarbodiimides.

# 3. ROTf-induced annulation of sulfur-containing substrates with alkynes

## 3.1. ROTf-induced annulation of arylisothiocyanates

The isothiocyanates possess the chemical group −N=C=S, which represents versatile reactivity in the synthesis of nitrogen- or sulfur-containing heterocycles. Comparatively speaking, the sulfur atom is strongly nucleophilic. Recently, we reported MeOTf as an electrophile [59] to react with aryisothiocyanates and alkynes leading to diverse highly substituted quinolones. The representative results are shown in scheme 8. A tandem electrophilic activation/cyclization via intermediate **I-18** is believed to be a possible process. This reaction demonstrated superiority on substrate scope for isothiocyanates. Furthermore, unsymmetrical alkynes such as terminal alkynes, (bromoethynyl) benzenes, even alkynes containing ester group could be applied in this reaction. In addition, alkyltriflates bearing C−C triple bond gave polycyclic quinolines **11d** via sequent cyclization process. Great effort has been paid in the transformation of a thioalkoxyl group such as oxidation, reduction and cross-coupling reaction, which make this method more powerful in organic synthesis [59]. This reaction represents a concise, metal-free and one-pot method for synthesis of functionalized quinolines.

## 3.2. ROTf-induced annulation of alkylisothiocyanates

Substitution of arylisothiocyanates with alkylisothiocyanates that lack an aryl group for the ring closure might lead to a new reaction mode. To our delight, the reaction proceeded well to afford indenone **12** after hydrolysis [60]. A range of arylalkynes could be employed in this reaction. The representative results are summarized in scheme 9. A plausible mechanism is also described in scheme 9. MeOTf as an electrophile reacts with isothiocyanate to form methylthio-substituted carbenium ion **I-19**, which followed the reaction with arylalkyne to form intermediate **I-20**. Utilization of the arylisothiocyanate affords quinoline **11**. Without aryl group in the alkylisothiocyanate, the nucleophilicity-strong sulfur atom attacks carbenium of **I-20** to form four-membered thiete **I-21**, which could be followed by ring opening with the C−S bond cleavage to form carbenium **I-23** via intermediate **I-22**. Finally, intramolecular Friedel−Crafts reaction of **I-23** affords

**Scheme 6.** ROTf-induced cyclization of diarylcarbodiimides and dialkylcarbodiimides.

indenone imine **I-24**, which undergoes hydrolysis to form indenone **12**. This reaction represents the first example of cleavage C−S bond in the isothiocyanate for construction of the carbocyclic compound under metal-free conditions.

More recently, Li and co-workers [61] reported MeOTf-induced intramolecular cyclization of isothiocyanates to afford 1-(methylthio)-3,4-dihydroisoquinolines **13** (scheme 10). The reaction may process by a tandem electrophilic activation and intramolecular Friedel−Crafts reaction.

# 4. ROTf-induced annulation of oxygen-containing substrates

## 4.1. ROTf-induced annulation of aldehydes

Apart from S- and N-reagents that could react with ROTf straightforwardly, O-reagents also demonstrated a good affinity with ROTf. As a part of ongoing projects on the alkyltriflate-triggered annulation, a reaction of MeOTf, aldehydes and arylalkynes was investigated [62] and a variety of 2,3-disubstituted 1-indanones was obtained. The representative results are shown in scheme 11. It is

**Scheme 7.** ROTf-induced cyclization of azobenzenes.

**Scheme 8.** ROTf-induced cyclization of arylisothiocyanates and arylalkynes.

noteworthy that a catalytic amount of MeOTf was employed and the reaction proceeded in satisfactory yield. Although the reaction mechanism is not clean, a plausible mechanism is shown at the bottom of scheme 11. First, MeOTf as an electrophile reacts with an aldehyde to afford the oxonium **I-25**, which couples with alkyne to form the highly active oxetenium intermediate **I-26** via [2 + 2] cycloaddition.

rsos.royalsocietypublishing.org    R. Soc. open sci. **5**: 181389

**Scheme 9.** ROTf-induced annulation of alkylisothiocyanates and arylalkynes.

**Scheme 10.** MeOTf-induced intramolecular cyclization of isothiocyanates.

Then, the intermediate **I-26** undergoes spontaneous isomerization to form the $4\pi$-Nazarov intermediate **I-27**, followed by Nazarov cyclization to give 1-indanone **14** and regeneration of MeOTf. This reaction provides a practical and convenient method for the synthesis of 2,3-disubstituted 1-indanones from readily available starting materials via MeOTf-induced catalysis.

**Scheme 11.** MeOTf-induced cyclization of aldehyde with arylalkynes.

## 4.2. ROTf-induced annulation of arylisocyanates

Isocyanate is the functional group with the formula R−N=C=O, in which N and O may both be alkylated by ROTf. We investigated a reaction of MeOTf, arylisocyanates and arylalkynes [63]. Notably, a range of 4-methoxyl-2,3-diarylquinolines **15** was obtained in good yields and the representative results are shown in scheme 12. Based on the results, a tandem [2 + 2] cycloaddition and intramolecular Friedel–Crafts reaction may be included in the reaction pathway. It is noteworthy that this reaction has limitations with only diarylalkynes and MeOTf for the construction of 4-methoxyl-2,3-diarylquinolines **15**.

Phenanthridinones are extensively found in natural products and bioactive molecules. We envisioned that MeOTf-induced intramolecular annulation of 2-phenyl aryllisocyanates would provide a pathway for the synthesis of phenanthridinones. During the course of our study on the $CO_2$ chemistry [64–67], we found a one-pot method for the synthesis of phenanthridinones [68] based on the MeOTf- and TBD-mediated carbonylation of *ortho*-arylanilines with $CO_2$. The representative results and reaction pathway are shown in scheme 13. This reaction shows MeOTf-induced carbonylation reaction of *o*-arylanilines applying $CO_2$ as the ideal carbonyl source to synthesize phenanthridinones containing a free (NH)-lactam motif under metal-free conditions.

## 5. ROTf-induced annulation of phosphorus-containing substrates

In 2006, Bates & Gates [69] used the strong electrophilicity of MeOTf in the synthesis of highly strained four-membered phosphorus heterocycles with phosphaalkenes as a precursor (scheme 14). The structure

**Scheme 12.** MeOTf-induced cyclization of arylisocyanates and arylalkynes.

**Scheme 13.** MeOTf-induced cyclization with *o*-arylanilines and CO₂.

**Scheme 14.** MeOTf-induced cyclization with phosphaalkenes.

of the unprecedented diphosphetanium salt **17** was identified by single crystal X-ray diffraction. Although the reaction mechanism is not clean, this reaction demonstrated a convenient method for synthesis of highly strained phosphorus heterocycles, which may be used as a propagating species in the cationic polymerization of phosphaalkenes.

rsos.royalsocietypublishing.org R. Soc. open sci. **5**: 181389

# 6. Conclusion and future outlook

ROTf has been a powerful reagent in organic synthesis featuring efficient, metal-free and easy of handing. A range of heteroatom-containing unsaturated reagents such as nitriles, carbodiimides, azobenzenes, isothiocyanates, aldehydes, isocyanates and phosphaalkene could be alkylated by the ROTf to generate reactive intermediates, which are capable of capturing other electrophilic substrates to afford cyclic compounds with a rational design. The strong electrophilicity of ROTf has bestowed the heteroatom-containing reagent quite unique versatility as a catalyst, promoter or reactant in a wide range of organic transformations, including carbon–carbon and carbon–heteroatom bond formation processes. The synthetic methodology to various carbocyclic and heterocyclic compounds has been widely developed. Although ROTf has exhibited great value for the synthesis of cyclic compounds, further exploration is required for the utilization of functionalized ROTf reagents and other heteroatom reagents such as organoselenium, organophosphorus substrates for synthesis of functionalized compounds. Furthermore, ROTf-induced unsaturated heteroatom-containing reagents to generate electrophiles bearing carbon cations could be involved in new approaches in many synthetic organic reactions under mild and metal-free conditions. Moreover, ROTf-induced $[2 + 2]$ cycloaddition to afford four-member ring intermediates proposed to elucidate surprising rearrangements still needs firm evidence. We anticipate that ROTf can be extended to more organic reactions in organic synthesis.

Data accessibility. This article does not contain any additional data.
Authors' contributions. S.Z. and S.W. participated in the design of the study and drafted the manuscript. C.X. conceived the study, coordinated the study and helped draft the manuscript. All authors gave final approval for publication.
Competing interests. We declare we have no competing interests.
Funding. This work was supported by the National Natural Science Foundation of China (nos. 21472106 and 91645120).
Acknowledgements. We thank all the cited authors for providing the research results for our reference.

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
