## [Reviewer comments · Royal Society Open Science]

Review History

RSOS-181005.R0 (Original submission)

Review form: Reviewer 1

Is the manuscript scientifically sound in its present form?

Yes

Are the interpretations and conclusions justified by the results?

Yes

Is the language acceptable?

No

Is it clear how to access all supporting data?

Not Applicable

Do you have any ethical concerns with this paper?

No

Have you any concerns about statistical analyses in this paper?

No

Recommendation?

Reject

Comments to the Author(s)

This review is basically a list of cyclization reactions brought about by treating various heteroatom-substituted compounds with $\text{MeOSO}_2\text{CF}_3$ in the presence of compounds containing triple or double bonds.

The quality of the English is poor and I am sending as an attachment a scan of a partially corrected copy (Appendix A).

My overall impression is that the presentation lacks much intellectual content--it is simply a list illustrated by exemplary equations. I think the work is more appropriate for a journal that specializes in heterocycles; the readers of such journals would have the motivation to examine the document. As far as I can make out from the on-line information about the journal the present type of review is outside the scope of the Royal Society Open Science Journal.

Review form: Reviewer 2

Is the manuscript scientifically sound in its present form?

No

Are the interpretations and conclusions justified by the results?

No

Is the language acceptable?

No

Is it clear how to access all supporting data?

Not Applicable

Do you have any ethical concerns with this paper?

No

Have you any concerns about statistical analyses in this paper?

No

Recommendation?

Reject

Comments to the Author(s)

The manuscript is a brief review of the triflate (ROTF)-induced reactions of heteroatom reagents and unsaturated substrates to afford cyclic compounds.

In general, it is poorly written and the standard of English is disappointing. There is no original science in the paper, since it is a review - was it an invited review?

The literature coverage is not sufficiently comprehensive to be a full review.

Taking all these factors into account, I cannot recommend acceptance.

Review form: Reviewer 3

Is the manuscript scientifically sound in its present form?

Yes

Are the interpretations and conclusions justified by the results?

Yes

Is the language acceptable?

No

Is it clear how to access all supporting data?

Not Applicable

Do you have any ethical concerns with this paper?

No

Have you any concerns about statistical analyses in this paper?

No

Recommendation?

Reject

Comments to the Author(s)

This review manuscript from Zou et al. gives an overview of the use of alkyltriflates for activation of cationic pathways to promote a number of cascade processes. This mini-review covers a number of transformations, which usually result in the formation of cyclic products, discussing aspects of the substrate scope, giving examples and highlighting probable mechanistic pathways.

Unfortunately, the manuscript is lacking somewhat in the introductory content; the background and prior art is not discussed in any great detail and the topic of the review is not surmised particularly well. There are also numerous errors in the schemes as well as the text. The work discussed within the review itself, albeit interesting, is also very niche. There are only a few examples presented and only 11 papers cited, the vast majority of which are from the authors own group. Hence, I do not find this work of sufficient value for publication in Royal Society Open Science.

Decision letter (RSOS-181005.R0)

03-Aug-2018

Dear Professor Xi:

Manuscript ID: RSOS-181005

Title: "RO1f-induced annulation of heteroatom reagents and unsaturated substrates leading to cyclic compounds"

Thank you for submitting the above manuscript to Royal Society Open Science. Your paper was sent to reviewers and their comments are included at the bottom of this letter.

In view of the concerns raised by the reviewers, the manuscript has been rejected in its current form. However, a new manuscript may be submitted which takes into consideration these comments.

Please note that resubmitting your manuscript does not guarantee eventual acceptance, and that your resubmission will be subject to peer review before a decision is made.

Your resubmitted manuscript should be submitted by 31-Jan-2019. If you are unable to submit by this date please contact the Editorial Office.

Yours sincerely,
Thomas Foley
Publishing Editor, Journals

Royal Society of Chemistry
Thomas Graham House
Science Park, Milton Road
Cambridge, CB4 0WF
Tel: +44 (0)1223 432516
Royal Society Open Science - Chemistry Editorial Office

On behalf of the Subject Editor Professor Anthony Stace and the Associate Editor Dr Andrew Harned

REVIEWER(S) REPORTS:

Associate Editor Comments to Author:

RSC Associate Editor: 1

Comments to the Author:

It is clear from the referee reports that the authors' presentation in this invited review has missed the mark. I agree with the referees that the language should be improved. However, I do disagree with the manner in which some of the criticisms were presented (particularly Referee 1). Nevertheless, the overall criticisms are valid.

As this was an invited review I would welcome a resubmission of a substantially revised manuscript that takes into account the criticisms raised by all three referees. Also, I strongly suggest modifying the text in such a way that it is more than just a list of what has been done. Are there obvious deficiencies in the literature that could be addressed by future researchers? Are there cases where these methods are dramatically better than alternatives? Where will we see the next advances in this area?

Reviewers' Comments to Author:

Reviewer: 1

Comments to the Author(s)

This review is basically a list of cyclization reactions brought about by treating various heteroatom-substituted compounds with $\text{MeOSO}_2\text{CF}_3$ in the presence of compounds containing triple or double bonds.

The quality of the English is poor and I am sending as an attachment a scan of a partially corrected copy.

My overall impression is that the presentation lacks much intellectual content--it is simply a list illustrated by exemplary equations. I think the work is more appropriate for a journal that specializes in heterocycles; the readers of such journals would have the motivation to examine the document. As far as I can make out from the on-line information about the journal the present type of review is outside the scope of the Royal Society Open Science Journal.

Reviewer: 2

Comments to the Author(s)

The manuscript is a brief review of the triflate (ROTf)-induced reactions of heteroatom reagents and unsaturated substrates to afford cyclic compounds.

In general, it is poorly written and the standard of English is disappointing. There is no original science in the paper, since it is a review - was it an invited review?

The literature coverage is not sufficiently comprehensive to be a full review.

Taking all these factors into account, I cannot recommend acceptance.

Reviewer: 3

Comments to the Author(s)

This review manuscript from Zou et al. gives an overview of the use of alkyltriflates for activation of cationic pathways to promote a number of cascade processes. This mini-review covers a number of transformations, which usually result in the formation of cyclic products, discussing aspects of the substrate scope, giving examples and highlighting probable mechanistic pathways.

Unfortunately, the manuscript is lacking somewhat in the introductory content; the background and prior art is not discussed in any great detail and the topic of the review is not surmised particularly well. There are also numerous errors in the schemes as well as the text. The work discussed within the review itself, albeit interesting, is also very niche. There are only a few examples presented and only 11 papers cited, the vast majority of which are from the authors

own group. Hence, I do not find this work of sufficient value for publication in Royal Society Open Science.

Author's Response to Decision Letter for (RSOS-181005.R0)

See Appendices B & C.

RSOS-181389.R0

Review form: Reviewer 3

Is the manuscript scientifically sound in its present form?

Yes

Are the interpretations and conclusions justified by the results?

Yes

Is the language acceptable?

Yes

Is it clear how to access all supporting data?

Not Applicable

Do you have any ethical concerns with this paper?

No

Have you any concerns about statistical analyses in this paper?

No

Recommendation?

Accept with minor revision (please list in comments)

Comments to the Author(s)

This resubmitted manuscript is significantly improved from its previous form and the authors have addressed the majority of the issues risen by the reviewers on the initial submission. I do think that the document is somewhat lacking in the conclusions section, it would be better to see more of a comparative summary which highlights the benefits of these methodologies over the current state-of-the-art. Why is this approach better? What unique reactivities are there? Does the methodology provide access to unique architectures? Where will these methodologies lead in the future?

As this is an invited review I am inclined to accept this minireview with the caveat that the authors expand somewhat upon their closing remarks.

Review form: Reviewer 4

Is the manuscript scientifically sound in its present form?

Yes

Are the interpretations and conclusions justified by the results?

Yes

Is the language acceptable?

Yes

Is it clear how to access all supporting data?

Yes

Do you have any ethical concerns with this paper?

No

Have you any concerns about statistical analyses in this paper?

No

Recommendation?

Accept as is

Comments to the Author(s)

This review manuscript entitled "ROTf-induced annulation of heteroatom reagents and unsaturated substrates leading to cyclic compounds" by Xi and coworkers gives an overview of the use of alkyltriflates for the synthesis of various cyclic compounds. Three reviewers have put forward some suggestions about the original manuscript. The authors have corrected the whole manuscript carefully. This reviewer recommend acceptance of this revised review as it.

Decision letter (RSOS-181389.R0)

09-Oct-2018

Dear Professor Xi:

Title: ROTf-induced annulation of heteroatom reagents and unsaturated substrates leading to cyclic compounds

Manuscript ID: RSOS-181389

Thank you for submitting the above manuscript to Royal Society Open Science. On behalf of the Editors and the Royal Society of Chemistry, I am pleased to inform you that your manuscript will be accepted for publication in Royal Society Open Science subject to minor revision in accordance with the referee suggestions. Please find the reviewers' comments at the end of this email.

The reviewers and handling editors have recommended publication, but also suggest some minor revisions to your manuscript. Therefore, I invite you to respond to the comments and revise your manuscript.

Please also include the following statements alongside the other end statements. As we cannot publish your manuscript without these end statements included, if you feel that a given heading is not relevant to your paper, please nevertheless include the heading and explicitly state that it is not relevant to your work. We have included a screenshot example of the end statements for reference.

- Ethics statement

Please clarify whether you received ethical approval from a local ethics committee to carry out your study. If so please include details of this, including the name of the committee that gave consent in a Research Ethics section after your main text. Please also clarify whether you received informed consent for the participants to participate in the study and state this in your Research Ethics section.

OR

Please clarify whether you obtained the necessary licences and approvals from your institutional animal ethics committee before conducting your research. Please provide details of these licences and approvals in an Animal Ethics section after your main text.

OR

Please clarify whether you obtained the appropriate permissions and licences to conduct the fieldwork detailed in your study. Please provide details of these in your methods section.

Because the schedule for publication is very tight, it is a condition of publication that you submit the revised version of your manuscript before 18-Oct-2018. Please note that the revision deadline will expire at 00.00am on this date. If you do not think you will be able to meet this date please let me know immediately.

- 1) A text file of the manuscript (tex, txt, rtf, docx or doc), references, tables (including captions) and figure captions. Do not upload a PDF as your "Main Document".
- 2) A separate electronic file of each figure (EPS or print-quality PDF preferred (either format should be produced directly from original creation package), or original software format)
- 3) Included a 100 word media summary of your paper when requested at submission. Please ensure you have entered correct contact details (email, institution and telephone) in your user account
- 4) Included the raw data to support the claims made in your paper. You can either include your data as electronic supplementary material or upload to a repository and include the relevant doi within your manuscript
- 5) All supplementary materials accompanying an accepted article will be treated as in their final form. Note that the Royal Society will neither edit nor typeset supplementary material and it will

be hosted as provided. Please ensure that the supplementary material includes the paper details where possible (authors, article title, journal name).

Best wishes,

Laura Smith, MRSC
Publishing Editor, Journals
Royal Society of Chemistry,
Thomas Graham House,
Science Park, Milton Road,
Cambridge, CB4 0WF, UK

Royal Society Open Science - Chemistry Editorial Office

On behalf of the Subject Editor Professor Anthony Stace and the Associate Editor Dr Andrew Harned.

RSC Associate Editor

Comments to the Author:

I agree with the reviewers that the revised manuscript is much improved over the previous version. At the same time, I agree with the suggestion that the conclusion and outlook could be expanded and improved upon.

Reviewer comments to Author:

Reviewer: 3

Comments to the Author(s)

This resubmitted manuscript is significantly improved from its previous form and the authors have addressed the majority of the issues risen by the reviewers on the initial submission. I do think that the document is somewhat lacking in the conclusions section, it would be better to see more of a comparative summary which highlights the benefits of these methodologies over the current state-of-the-art. Why is this approach better? What unique reactivities are there? Does the methodology provide access to unique architectures? Where will these methodologies lead in the future?

As this is an invited review I am inclined to accept this minireview with the caveat that the authors expand somewhat upon their closing remarks.

Reviewer: 4

Comments to the Author(s)

This review manuscript entitled "ROTf-induced annulation of heteroatom reagents and unsaturated substrates leading to cyclic compounds" by Xi and coworkers gives an overview of the use of alkyltriflates for the synthesis of various cyclic compounds. Three reviewers have put forward some suggestions about the original manuscript. The authors have corrected the whole manuscript carefully. This reviewer recommend acceptance of this revised review as it.

Author's Response to Decision Letter for (RSOS-181389.R0)

See Appendix D.

Decision letter (RSOS-181389.R1)

16-Oct-2018

Dear Professor Xi:

Title: ROTf-induced annulation of heteroatom reagents and unsaturated substrates leading to cyclic compounds

Manuscript ID: RSOS-181389.R1

It is a pleasure to accept your manuscript in its current form for publication in Royal Society Open Science. The chemistry content of Royal Society Open Science is published in collaboration with the Royal Society of Chemistry.

On behalf of the Subject Editor Professor Anthony Stace and the Associate Editor Dr Andrew Harned.

RSC Associate Editor
Comments to the Author:
(There are no comments.)

Reviewer(s)' Comments to Author:

ROYAL SOCIETY
OPEN SCIENCE

R. Soc. open sci.
doi:10.1098/not yet assigned

**ROTf-induced annulation of heteroatom reagents and unsaturated substrates
leading to cyclic compounds**

Song Zou[#], Sheng Wang[#], Chanjuan Xi^{*}

MOE Key Laboratory of Bioorganic Phosphorus Chemistry & Chemical Biology, Department of
Chemistry, Tsinghua University, Beijing 100084, China

Keywords: trifluoromethanesulfonates; heteroatom reagents; annulation; cyclic compounds

1. Summary

The development of metal-free organic reactions is one of the hotspots in the synthesis of cyclic compounds. ROTf (alkyl trifluoromethanesulfonates), due to their good electrophilicity are powerful alkylated reagents to unsaturated heteroatom reagents leading to carbon cation, which could subsequently reacted with appropriate substrates to produce cyclic compounds. In this review, we mainly focus on the ROTf-induced annulation of unsaturated heteroatom reagents such as nitriles, carbodiimides, azobenzenes, isothiocyanates, aldehydes, isocyanates, and phosphalkene with themselves or alkynes to afford cyclic compounds.

*Author for correspondence (cjxi@tsinghua.edu.cn).

[#]Present address: Department of Chemistry, Tsinghua University, Beijing 100084, China.

NOT CLEAR!
WHAT IS MEANT -

2. Introduction

ROTf (alkyl trifluoromethanesulfonates) are powerful alkylation reagents *which are* and frequently utilized in alkylation of nucleophiles [1]. Benefit from fluorine in ^-OTf to stabilize the negative charge and possessing excellent leaving ability, which make the ROTf much more reactive as alkylation reagents than alkyl iodide or $(MeO)_2SO_2$. As results of extensive application in organic synthesis, a lot of methods have been developed to prepare ROTf, such as reaction of orthoformates and triflic anhydrides under solvent-free condition [2]. The ROTf were often used as alkylation reagents for diverse substrates containing heteroatom^s such as P [3], O [4], N [5-7], S [8,9], Te [10], Ge [11], Bi [12], Si [13], and Se [14,15], which straightforwardly afforded the corresponding alkylated products with aid of base or stable triflates irreversibly that even could be applied as ionic liquids [16,17]. Moreover, the resulted ^{ing}triflates could be further transformed to other products by substitution [18] (Scheme 1, a). On the other hand, unsaturated heteroatom reagents, such as nitriles, aldehydes, isocyanates, and isothiocyanates, were captured by ROTf to generate electrophiles bearing carbon cation [19], which could be subsequently reacted with appropriate unsaturated substrates to produce cyclic compounds (Scheme 1, b). ROTf-induced cyclization featuring metal-free, easy to handle, and good region^s selectivity provided a feasible approach to diverse cyclic compounds. In this review, we mainly focus on the ROTf-induced annulation of unsaturated heteroatom reagents such as nitriles, carbodiimides, azobenzenes, isothiocyanates, aldehydes, isocyanates, and phosphalkene with themselves or alkynes to afford cyclic compounds.

a. ROTf-mediated substitution:

b. ROTf-mediated cyclization:

Scheme 1. ROTf-mediated reaction.

3. ROTf-induced annulation of nitrogen-containing substrates with unsaturated compounds

3.1 ROTf-induced annulation of nitriles with alkynes

Nitriles as unsaturated heteroatom reagents could react with ROTf to form *N*-alkylated nitriliums, which were well investigated by Booth [7] in 1980. However, electrophilicity of *N*-alkylated nitriliums has not been utilized in further reaction until recent ~~year~~^{ly}. In 2014, our group [20] reported MeOTf-induced carboannulation of aryl nitriles and aromatic alkynes to construct indenones **1** (Scheme 2). Triflate **3** was isolated when dec-5-yne was used indicating **I-1** might be an intermediate in this reaction. A range of functionalized indenone products were obtained. When *ortho*-substituted aryl nitriles were utilized, indenone imine **I-1** would further cyclize with another molecular of nitriliums to give indeno [1,2-*c*]-isoquinolines **2** with construction of one carbocycle and one heterocycle.

Scheme 2. MeOTf-induced cyclization of aryl nitriles and alkynes.

Aryl nitriles and alkynes could be induced by MeOTf to afford indenones via intermediate I-5. We envisioned the utilization of alkylnitriles, which lacks an aryl group for the ring closure, might lead to a different ring formation process. To our delight, reaction of alkylnitriles, alkynes, and MeOTf indeed afforded tetrasubstituted NH-pyrroles **4** in high regioselectivity, which includes one carbon from MeOTf to join the final products [21] (Scheme 3). It is noteworthy that the cyclized pyrrole captures another nitrilium leading to highly substituted NH-pyrroles after hydrolysis. When EtOTf was utilized instead of MeOTf, the product **5** was obtained.

Scheme 3. MeOTf-induced cyclization of alkylnitriles and alkynes.

Furthermore, when 1,2-diphenylethyne and 1,2-di-*p*-tolylethyne were employed at 130 °C, the isoquinolines **6** were obtained in good yields. The representative results are summarized in Scheme 4.

Scheme 4. MeOTf-induced cyclization of alkylnitriles and diarylalkynes

3.2 ROTf-induced annulation of carbodiimides

31
32
33
34
35
36
37
38
39
40
41
42
43
44
45
46
47
48
49
50
51
52
53
54
55
56
57
58
59
60

More recently, ROTf-induced electrophilic cyclization was extended to carbodiimides. Our group demonstrated a ROTf-triggered intermolecular cyclization of carbodiimides to afford a range of 2-amino-4-imino-quinazolines **9**, 2,4-diaminoquinazolines **7** and 2-aminoquinazolinones **8** efficiently, which are important motifs in pharmaceuticals [22]. When the *N,N'*-diarylcarbodiimides were employed, the reaction proceeded smoothly to afford the corresponding 2-amino-4-imino-quinazolines **9**, which could be hydrolyzed to generate the corresponding 2-aminoquinazolinones **8**. The representative results are shown in Scheme 5.

R. Soc. open sci. article template

7

Scheme 5. MeOTf-induced cyclization of diarylcarbodiimides

To further extend the substrate scope, a combination of two different carbodiimides has been achieved. A range of diarylcarbodiimides and dialkylcarbodiimides ^{was} investigated under the optimized the reaction condition. The representative results are shown in Scheme 6. A plausible mechanism was proposed. First, ^{the} carbodiimide is methylated by MeOTf and subsequently attacked by another molecule ^{of} carbodiimide to give intermediate **I-11**. Then, intramolecular nucleophilic attack takes place to afford ^{the} four-membered intermediate **I-12**, which

WHERE IS DIAG I?

generates carbenium **7** after ring opening via C–N bond cleavage. Finally, intramolecular Friedel-Crafts annulation occurs to form the corresponding quinazolinone imine, which could give 2, 4-diamino-quinazoline **9** (R = alkyl group) and 2-amino-quinazolinone **8** (R = aryl group) after hydrolysis.

Scheme 6. ROTf-induced cyclization of diarylcarbodiimides and dialkylcarbodiimides.

3.3 ROTf-induced annulation of azobenzenes

R. Soc. open sci. article template

9

Apart from nitriles and carbodiimides, azobenzenes are also significant nitrogen-containing compounds, which possess Lewis basicity. In 2014, our group demonstrated MeOTf-induced cyclization of azobenzenes by N=N bond cleavage with aid of TCQ (tetrachloro-1,4-benzoquinone) as oxidant to afford *N*-arylbenzimidazoles **10** [23] (Scheme 7). When unsymmetrical azobenzenes were used, cyclization tends to occur ^{on the} ~~in~~ electron-rich anisoyl ring. EtOTf could facilitate N=N bond cleavage as well to generate 2-methylbenzimidazole **10d**. ^{AP} Plausible mechanism ^{is} ~~was demonstrated~~ as shown in Scheme 7.

Scheme 7. MeOTf-induced cyclization of azobenzenes.

4. ROTf-induced annulation of sulfur-containing substrates

4.1 ROTf-induced annulation of arylisothiocyanates with alkynes

Isothiocyanates are widely used as reagents in organic synthesis ^{possessing} ~~possessing~~ functional group $N=C=S$, in which sulfur ^{The} ~~is~~ ^{by} ~~is~~ a heteroatom is strong nucleophilic ~~ly~~.

Recently, we reported MeOTf as an electrophile [24] to react with arylisothiocyanates and alkynes leading to diverse highly substituted quinolones (Scheme 8). Tandem electrophilic cyclization *via* intermediate **I-18** is believed to be a possible process. This reaction demonstrated superiority on substrate scope. Unsymmetrical alkynes such as terminal alkynes, (bromoethynyl) benzenes, even alkynes containing ester group could be applied in this reaction. Furthermore, alkyltriflates bearing C-C triple bond gave polycyclic quinolines **11d** *via* bicyclization process. Great effort has been paid in transformation of thioalkoxy group such as oxidation, reduction, and cross-coupling reaction, which make this method more powerful in organic synthesis.

Scheme 8. ROTf-induced cyclization of arylisothiocyanates and arylalkynes.

4.2 ROTf-induced annulation of alkyisothiocyanates

Substitution of arylisothiocyanates with alkyisothiocyanate gives a new reaction mode [25] with formation of indenones (Scheme 9). A four-membered ring **I-21** was included in the reaction pathway. Moreover, transformation of thioalkoxy group at the 3-position of the indenone to phenyl, methylsulfonyl, amino, and ethoxy groups was explored as well, exhibiting potential utilization in preparation of highly substituted indenones.

R. Soc. open sci. article template

11

Scheme 9. ROTf-induced annulation of alkylisothiocyanates and arylalkynes.

More recently, Li and co-workers [26] reported MeOTf-induced intramolecular cyclization of isothiocyanates to afford thioisoquinolines **13** (Scheme 10).

Scheme 10. MeOTf-induced intramolecular cyclization of isothiocyanates.

5. ROTf-induced annulation of oxygen-containing substrates

5.1 ROTf-induced annulation of aldehydes

?
↑
WRONG NAME
DIHYDRO?

1
2
3
4 Apart from S and N which could react with ROTf straightforwardly, O also
5 demonstrated good affinity with ROTf. As a part of ongoing projects on the
6 alkyltriflate-triggered annulation, our group [27] investigated a reaction of MeOTf,
7 aldehydes, and arylalkynes (Scheme 11). A variety of 2,3-disubstituted 1-indanones
8 were obtained. It is noteworthy that ^{ly} catalytic amount of MeOTf was employed and the
9 reaction proceeded in satisfactory yield. We believe that MeOTf catalyzed the
10 annulation of aldehydes and arylalkynes to afford 2, 3-disubstituted indanones **14** by
11 [2 + 2] cycloaddition and Nazarov cyclization in sequence as shown in Scheme 11.
12
13
14
15
16
17
18

I-29 → I-27
SOMETHING
WRONG
HERE

Scheme 11. MeOTf-induced cyclization of aldehyde with arylalkynes.

5.2 ROTf-induced annulation of arylisocyanates

44 Isocyanate is the functional group with the formula R-N=C=O, in which N and
45 O both may ^{both be} alkylated by ROTf. On the basis of our former research works ^{ly} on
46 alkyltriflate-triggered annulation reactions, our group [28] investigated a reaction of
47 MeOTf, arylisocyanates, and arylalkynes (Scheme 12). Notably, a range of 4-
48 methoxyl-2,3-diarylquinolines was obtained in good yields. A plausible reaction
49 pathway is shown in Scheme 12, and a tandem [2 + 2] and intramolecular
50 Friedel-Crafts reaction is included.
51
52
53
54
55
56
57
58
59
60

R. Soc. open sci. article template

13

Scheme 12. MeOTf-induced cyclization of arylisocyanates and arylalkynes.

Phenanthridinones are extensively ^{found} exist in natural products and bioactive molecules. We envisioned that MeOTf-induced intramolecular annulation of 2-phenyl aryllisocyanates would provide a pathway for the synthesis of phenanthridinones. During the course of our study on the CO₂ chemistry, we found a one-pot method for the synthesis of phenanthridinones [29] based on the MeOTf- and TBD-mediated carbonylation of *ortho*-arylanilines with CO₂. The representative results and reaction pathway are shown in Scheme 13.

Scheme 13. MeOTf-induced cyclization with *o*-arylanilines and CO₂.

6. ROTf-induced annulation of phosphorus-containing substrates

In 2006, Gates and co-workers [30] utilized ^{the} strong electrophilicity of MeOTf in ^{the} synthesis of highly strained four-membered phosphorus heterocycles with phosphalkenes as precursor (Scheme 14). ^{The} Structure of ^{the} unprecedented diphosphetanium salt **17** was identified by single crystal X-ray diffraction.

Scheme 14. MeOTf-induced cyclization with phosphalkenes.

7. Conclusion and future outlook

ROTf has been a powerful reagent in organic synthesis featuring efficient, metal-free, and ^{easy to handling} easy-handled reagents. A range of heteroatom-containing unsaturated reagents such as nitriles, carbodiimides, azobenzenes, isothiocyanates, aldehydes, isocyanates, and phosphalkene could be alkylated by the ROTf to generate reactive intermediates,

R. Soc. open sci. article template

15

1
2
3
4 which are capable of capturing other substrates to afford cyclic compounds with
5 rational design. The synthetic methodology to various carbocyclic and heterocyclic
6 compounds ~~is~~ ^{has, been} widely developed. Although ~~the~~ ^{is} ROTf has exhibited great value for the
7 synthesis of cyclic compounds, further exploration is required for the utilization of
8 functionalized ROTf reagents and other heteratom reagents such as organoselenium,
9 organophosphorus ^{compounds}. Furthermore, four-member ring intermediates proposed to
10 elucidate surprising rearrangement ~~is still lack of~~ ^{s needs} firm evidence. We anticipate that
11 ROTf can be extended to more organic reactions in organic synthesis.
12
13
14
15
16
17
18

19 Data accessibility

20 As a review article, there is no original data included.
21

22 Acknowledgements

23 We thank all the cited authors for providing the research results for our reference.
24
25
26

27 Funding

28 This work was supported by the National Natural Science Foundation of China (nos. 21472106
29 and 91645120).
30
31
32

33 Competing Interests

34 We declare we have no competing interests.
35
36
37

38 Authors' Contributions

39 S. Zou and S. Wang participated in the design of the study and drafted the
40 manuscript; [#]S. Zou and S. Wang contributed equally; C. Xi conceived the study,
41 coordinated the study and helped draft the manuscript. All authors gave final approval
42 for publication.
43
44
45
46
47
48
49
50
51
52
53
54
55
56
57
58
59
60

References

1. (a) Alder RW, Phillips JGE, Huang L, Huang X. 2005 "Methyltrifluoromethanesulfonate", *encyclopedia of reagents for organic synthesis*, John Wiley & Sons. (doi:10.1002/047084289X.rm266m.pub2); (b) Stang PJ, Hanack M, Subramanian LR. 1982 Perfluoroalkanesulfonic esters: methods of preparation and applications in organic chemistry. *Synthesis*. **2**, 0085-0127. (doi: 10.1055/s-1982-29711); (c) Howells RD, Mc Cown JD. 1977 Trifluoromethanesulfonic acid and derivatives. *Chem. Rev.* **77**, 69-92. (doi:10.1021/cr60305a005)
2. (a) Zheng D, Ma H, Ding K. 2017 A practical synthesis of trifluoromethanesulfonate esters. *Chin. J. Org. Chem.* **37**, 1582-1584. (doi: 10.6023/cjoc201701035); (b) Chapman RD, Andreshak JL, Herrlinger SP, Shackelford SA, Hildreth RA, Smith JP. 1986 Trifluoromethanesulfonate esters from dibromoalkane metatheses with silver triflate: mechanistic and synthetic aspects. *J. Org. Chem.* **51**, 3792-3798. (doi:10.1021/jo00370a009); (c) Baraznenok IL, Nenajdenko VG, Balenkova ES. 2000 Chemical transformations induced by triflic anhydride. *Tetrahedron*. **56**, 3077-3119. (doi:10.1016/S0040-4020(00)00093-4); (d) Aubert C, Begue JP. 1985 A simple preparation of alkyl trifluoromethanesulfonates (triflates) from alkyl trimethylsilyl ethers. *Synthesis*. **8**, 759-760. (doi:10.1055/s-1985-31336); (e) Burdon J, McLoughlin VCR. 1965 Trifluoromethanesulphonate esters and their alkylating properties. *Tetrahedron*. **21**, 1-4. (doi:10.1016/S0040-4020(01)82194-3); (f) Dobbs AP, Jones K, Veal KT. 1997 The generation and cyclisation of pyridinium radicals as a potential route to indolizidine alkaloids. *Tetrahedron Lett.* **38**, 5383-5386. (doi: 10.1016/S0040-4039(97)01178-7); (g) Beard CD, Baum K, Grakauskas V. 1973 Synthesis of some novel trifluoromethanesulfonates and their reactions with alcohols. *J. Org. Chem.* **38**, 3673-3677. (doi:10.1021/jo00961a003); (h) Ignatyev NV, Barthen P, Kucheryna A, Willner H, Sartori P. 2012 A method for producing perfluoroalkanesulfonic acid esters. *Molecules*. **17**, 5319-5337.

R. Soc. open sci. article template

17

- (doi:10.3390/molecules17055319)
3. Burford N, Dyker CA, Decken A. 2005 Facile synthetic methods for the diversification of catena-polyphosphorus cations. *Angew. Chem. Int. Ed.* **44**, 2364-2367. (doi:10.1002/anie.200462997)
 4. Imamoto T, Kikuchi S-I, Miura T, Wada Y. 2001 Stereospecific reduction of phosphine oxides to phosphines by the use of a methylation reagent and lithium aluminum hydride. *Org. Lett.* **3**, 87-90. (doi: 10.1021/ol0068041)
 5. Rong MK, Duin K, Dijk T, Pater JJM, Deelman B.-J, Nieger M, Ehlers AW, Slootweg JC, Lammertsma K. 2017 Iminophosphanes: synthesis, rhodium complexes, and ruthenium(II)-catalyzed hydration of nitriles. *Organometallics.* **36**, 1079-1090. (doi: 10.1021/acs.organomet.7b00057)
 6. Ye H, Liu R, Li D, Liu Y, Yuan H, Guo W, Zhou L, Cao X, Tian H, Shen J, Wang P. 2013 A safe and facile route to imidazole-1-sulfonyl azide as a diazotransfer reagent. *Org. Lett.* **15**, 18-21. (doi:10.1021/ol3028708)
 7. Booth BL, Jibodu KO, Proenca MF. 1980 The synthesis and some reactions of N-methylnitrilium trifluoromethanesulphonate salts. *J. Chem. Soc., Chem. Commun.* **0**, 1151-1153. (doi:10.1039/C39800001151)
 8. Thomas Z, Gregor L. 1998 Synthesis of β -mannosides via prearranged glycosides. *Angew. Chem. Int. Ed.* **37**, 3129-3132. (doi:10.1002/(SICI)1521-3773(19981204)37:22<3129::AID-ANIE3129>3.0.CO;2-8)
 9. Ranade SC, Kaothip S, Demchenko AV. 2010 Glycosyl alkoxythioimidates as complementary building blocks for chemical glycosylation. *Org. Lett.* **12**, 5628-5631. (doi:10.1021/ol1023079)
 10. Laali K, Chen HY, Gerzina RJ. 1988 Methylation of aromatics with Me_3S^+ , Me_3Se^+ , and Me_3Te^+ in superacid media. *J. Org. Chem.* **348**, 199-204. (doi:10.1016/0022-328X(88)80396-6)
 11. Su B, Ganguly R, Li Y, Kinjo R. 2016 Synthesis, characterization, and electronic structures of a methyl germyliumylidene ion and germylone-group VI metal complexes. *Chem. Commun.* **52**, 613-616. (doi:10.1039/C5CC08665E)

R. Soc. open sci.

<https://mc.manuscriptcentral.com/rsos>

- 1
2
3
4
5
6
7
8
9
10
11
12
13
14
15
16
17
18
19
20
21
22
23
24
25
26
27
28
29
30
31
32
33
34
35
36
37
38
39
40
41
42
43
44
45
46
47
48
49
50
51
52
53
54
55
56
57
12. Wallenhauer S, Seppelt K. 1994 Methylbismuth(V) compounds. *Angew. Chem. Int. Ed.* **33**, 976-978. (doi:10.1002/anie.199409761)
13. Yamaguchi T, Asay M, Sekiguchi A. 2012 $[(\text{Me}_3\text{Si})_2\text{CH}]_2^i\text{PrSi}(\text{NHC})\text{Si}=\text{Si}(\text{Me})\text{Si}^i\text{Pr}[\text{CH}(\text{SiMe}_3)_2]_2^+$: A molecule with disilyl cation character. *J. Am. Chem. Soc.* **134**, 886-889. (doi:10.1021/ja210669n)
14. Spera ML, Harman WD. 1999 Osmium-mediated electrophilic addition reactions with selenophene and activation of the Se-C bond. *Organometallics.* **18**, 1559-1561. (doi:10.1021/om9806949)
15. Mutoh Y, Murai T. 2003 Acyclic selenoiminium salts: isolation, first structural characterization, and reactions. *Org. Lett.* **5**, 1361-1364. (doi:10.1021/ol034334f)
16. Chen Z.-J, Xi H.-W, Lim KH, Lee J.-M. 2013 Distillable ionic liquids: reversible amide O alkylation. *Angew. Chem. Int. Ed.* **52**, 13392-13396. (doi:10.1002/anie.201306476)
17. Li G, Xue Z, Cao B, Yan C, Mu T. 2016 Preparation and properties of C=X (X: O, N, S) based distillable ionic liquids and their application for rare earth separation. *ACS. Sustain. Chem. Eng.* **4**, 6258-6262. (doi:10.1021/acssuschemeng.6b01984)
18. (a) Ye H, Liu R, Li D, Liu Y, Yuan H, Guo W, Zhou L, Cao X, Tian H, Shen J, Wang P. 2013 Organic letters: celebrating 15 years. *Org. Lett.* **15**, 1-2. (doi:10.1021/ol3033847); (b) Cordone R, Harman WD, Taube H. 1989 Carbon-hydrogen bond activation in novel eta-2-bound cationic heterocycle complexes of pentaammineosmium(II). *J. Am. Chem. Soc.* **111**, 2896-2900. (doi:10.1021/ja00190a025); (c) Ulibarri G, Choret N, Bigg DCH. 1996 Activation of imidazolides using methyl trifluoromethanesulfonate: a convenient method for the preparation of hindered esters and amides. *Synthesis.* **11**, 1286-1288. (doi:10.1055/s-1996-4399); (d) Zhu F, Tao J, Wang Z. 2015 Palladium-catalyzed C-H arylation of (benzo)oxazoles or (benzo)thiazoles with aryltrimethylammonium triflates. *Org. Lett.* **17**, 4926-4929. (doi:10.1021/acs.orglett.5b02458); (e) Yu W, Yang S, Xiong F, Fan T, Feng Y,

R. Soc. open sci. article template

19

- Huang Y, Fu J, Wang T. 2018 Palladium-catalyzed carbonylation of benzylic ammonium salts to amides and esters via C–N bond activation. *Org. Biomol. Chem.* **16**, 3099-3103. (doi:10.1039/C8OB00488A); (f) Ranade SC, Kaeothip S, Demchenko AV. 2010 Synthesis of click bile acid polymers and their application in stabilization of silver nanoparticles showing iodide sensing property. *Org. Lett.* **12**, 24-27. (doi:10.1021/ol902351g)
19. (a) Dondoni A, Catozzi N, Marra A. 2005 Concise and practical synthesis of α -glycosyl ketones from sugar benzothiazoles and their transformation into chiral tertiary alcohols. *J. Org. Chem.* **70**, 9257-9268. (doi:10.1021/jo051377w); (b) Rong MK, Duin KV, Dijk TV, Pater JJMD, Deelman BJ, Nieger M, Ehlers AW, Slootweg JC, Lammertsma K. 2017 Iminophosphanes: synthesis, rhodium complexes, and ruthenium(II)-catalyzed hydration of nitriles. *Organometallics.* **36**, 1079-1090. (doi:10.1021/acs.organomet.7b00057); (c) Imamoto T, Kikuchi S, Miura T Wada YY. 2001 Stereospecific reduction of phosphine oxides to phosphines by the use of a methylation reagent and lithium aluminumhydride. *Org. Lett.* **3**, 1-3. (doi:10.1021/ol006590n); (d) Murai T, Mutoh Y, Ohta Y, Murakami M. 2004 Synthesis of tertiary propargylamines by sequential reactions of in situ generated thioiminium salts with organolithium and -magnesium reagents. *J. Am. Chem. Soc.* **126**, 5968-5969. (doi:10.1021/ja048627v); (e) Mutoh Y, Murai T. 2003 Acyclic selenoiminium salts: isolation, first structural characterization, and reactions. *Org. Lett.* **5**, 1361-1364. (doi:10.1021/ol034334f)
20. Yan X, Zou S, Zhao P, Xi C. 2014 MeOTf-induced carboannulation of aryl nitriles and aromatic alkynes: a new metal-free strategy to construct indenones. *Chem. Commun.* **50**, 2775-2777. (doi:10.1039/C4CC00088A)
21. Liu Y, Yi X, Luo X, Xi C. 2017 MeOTf-mediated annulation of alkylnitriles and arylalkynes leading to polysubstituted NH-pyrroles. *J. Org. Chem.* **82**, 11391-11398. (doi:10.1021/acs.joc.7b01845)
22. Zhang X, Wang S, Liu Y, Xi C. 2018 Triflates-triggered intermolecular cyclization of carbodiimides leading to 2-aminoquinazolinone and 2,4-

- 1
2
3 diaminoquinazoline derivatives. *Org. Lett.* **20**, 2148-2151.
4 (doi:10.1021/acs.orglett.8b00314)
5
6
7 23. Yan X, Yi X, Xi C. 2014 Direct cleavage of the NN bond of azobenzenes by
8 MeOTf leading to N-arylbenzimidazoles. *Org. Chem. Front.* **1**, 657-660.
9 (doi:10.1039/C4QO00056K)
10
11
12 24. Zhao P, Yan X, Yin H, Xi C. 2014 Alkyltriflate-triggered annulation of
13 arylisothiocyanates and alkynes leading to multiply substituted quinolines through
14 domino electrophilic activation. *Org. Lett.* **16**, 1120-1123.
15 (doi:10.1021/ol500221u)
16
17
18 25. Zhao P, Liu Y, Xi C. 2015 MeOTf-induced carboannulation of isothiocyanates
19 and aryl alkynes with C=S bond cleavage: access to indenones. *Org. Lett.* **17**,
20 4388-4391. (doi:10.1021/acs.orglett.5b02201)
21
22
23 26. Wen L, Dou Q, Wang Y, Zhang J, Guo W, Li M. 2017 Synthesis of 1-thio-
24 substituted isoquinoline derivatives by tandem cyclization of isothiocyanates. *J.*
25 *Org. Chem.* **82**, 1428-1436. (doi:10.1021/acs.joc.6b02605)
26
27
28 27. Liu Y, Zhao P, Zhang B, Xi C. 2016 MeOTf-catalyzed annulation of aldehydes
29 and arylalkynes leading to 2,3-disubstituted indanones. *Org. Chem. Front.* **3**,
30 1116-1119. (doi:10.1039/C6QO00253F)
31
32
33 28. Liu Y, Zhang X, Xi C. 2018 MeOTf-induced annulation of arylisocyanates and
34 arylalkynes leading to 4-methoxyl-2,3-diarylquinolines. *Tetrahedron Lett.* **25**,
35 2440-2442. (doi: 10.1016/j.tetlet.2018.05.030)
36
37
38 29. Wang S, Shao P, Du G, Xi C. 2016 MeOTf- and TBD-mediated carbonylation of
39 ortho-arylanilines with CO₂ leading to phenanthridinones. *J. Org. Chem.* **81**,
40 6672-6676. (doi:10.1021/acs.joc.6b01318)
41
42
43 30. Bates JI, Gates PD, 2006 Diphosphiranium (P₂C) or diphosphetanium (P₂C₂)
44 cyclic cations: different fates for the electrophile-initiated cyclodimerization of a
45 phosphalkene. *J. Am. Chem. Soc.* **128**, 15998-15999. (doi:10.1021/ja0667662)
46
47
48
49
50
51
52
53
54
55
56
57

Appendix B

Respond to Editor and Referees

Respond to the RSC Associate Editor 1

Comments to the Author:

It is clear from the referee reports that the authors' presentation in this invited review has missed the mark. I agree with the referees that the language should be improved. However, I do disagree with the manner in which some of the criticisms were presented (particularly Referee 1). Nevertheless, the overall criticisms are valid.

As this was an invited review I would welcome a resubmission of a substantially revised manuscript that takes into account the criticisms raised by all three referees. Also, I strongly suggest modifying the text in such a way that it is more than just a list of what has been done. Are there obvious deficiencies in the literature that could be addressed by future researchers? Are there cases where these methods are dramatically better than alternatives? Where will we see the next advances in this area?

Answer:

- 1. We reorganized the manuscript and addressed ROTf-induced annulation as metal-free methods for synthesis of heterocycles. In addition, a variety of relative literatures has been discussed and reaction pathways have been described.**
- 2. Transition-metal-catalyzed reactions are some of the most attractive methodologies for synthesizing heterocycles, since a transition-metal-catalyzed reaction can directly construct complicated molecules from readily accessible starting materials under mild conditions. Nevertheless, transition-metal-catalyzed reactions are still limited in applications and confront challenges to some extent, since transition-metals are expensive, toxic, inconvenient for operation, and environment damage. In this regard, ROTf-induced cyclization featuring metal-free,**

easy to handle, and good selectivity provided a feasible approach to diverse cyclic compounds. We mentioned them in the introduction part.

- 3. Although ROTf-induced annulation as metal-free methods for synthesis of heterocycles has been summarized, further exploration is required for the utilization of functionalized ROTf reagents and other heteroatom reagents such as organoselenium, organophosphorus substrates for synthesis of functionalized compounds. Furthermore, four-member ring intermediates proposed to elucidate surprising rearrangements still needs firm evidence. We mentioned them in the conclusion and outlook's part.**

Respond to the Reviewer 1

Comments to the Author(s)

This review is basically a list of cyclization reactions brought about by treating various heteroatom-substituted compounds with MeOSO₃CF₃ in the presence of compounds containing triple or double bonds.

The quality of the English is poor and I am sending as an attachment a scan of a partially corrected copy.

My overall impression is that the presentation lacks much intellectual content-it is simply a list illustrated by exemplary equations. I think the work is more appropriate for a journal that specializes in heterocycles; the readers of such journals would have the motivation to examine the document. As far as I can make out from the on-line information about the journal the present type of review is outside the scope of the Royal Society Open Science Journal.

Answer:

- 1. We reorganized the manuscript and addressed ROTf-induced annulation as metal-free methods for synthesis of heterocycles. ROTf (alkyl trifluoromethanesulfonates) are powerful**

alkylating reagents for diverse substrates containing heteroatoms. The resulting triflates could be further transformed to other products by substitution/addition/cyclization. In this review, we mainly classified the typical ROTf-induced annulation of N-reagents, S-reagents, O-reagents and P-reagents with themselves or alkynes to afford cyclic compounds as well as summarized their recent advances from different aspects.

2. Thank Reviewer 1 to correct language, we reorganized the manuscript and all the grammatical errors and typographical errors were corrected according to the reviewer's suggestions.

Respond to the Reviewer 2

Comments to the Author(s)

The manuscript is a brief review of the triflate (ROTf)-induced reactions of heteroatom reagents and unsaturated substrates to afford cyclic compounds. In general, it is poorly written and the standard of English is disappointing. There is no original science in the paper, since it is a review - was it an invited review?

The literature coverage is not sufficiently comprehensive to be a full review.

Taking all these factors into account, I cannot recommend acceptance.

Answer:

1. We reorganized the manuscript and the English proof reading was carefully done with the assistance of a native English speaker.
2. Heterocycles are ubiquitous in natural products, pharmaceuticals, organic materials, and numerous functional molecules. Therefore, organic chemists have been making extensive efforts to produce these heterocyclic compounds by

developing new and efficient synthetic transformations. Recently, transition-metal-catalyzed reactions are one of the attractive methodologies for synthesizing heterocycles, however, transition-metal-catalyzed reactions are still limited in applications and confront challenges to some extent, since transition-metals are expensive, toxic, inconvenient for operation, and environment damage. In this regard, a transition metal-free methodology for the construction of an important heterocyclic compounds in drug discovery and material science has attracted attention. ROTf (alkyl trifluoromethanesulfonates) are powerful alkylating reagents for diverse substrates containing heteroatoms. The resulting triflates could be further transformed to other products by substitution/addition/cyclization. In this review, we mainly classified the typical ROTf-induced annulation of N-reagents, S-reagents, O-reagents and P-reagents with themselves or alkynes to afford cyclic compounds as well as summarized their recent advances from different aspects.

3. We carefully checked the literature and more references were added appropriately.

Respond to the Reviewer 3

Comments to the Author(s)

This review manuscript from Zou et al. gives an overview of the use of alkyltriflates for activation of cationic pathways to promote a number of cascade processes. This mini-review covers a number of transformations, which usually result in the formation

of cyclic products, discussing aspects of the substrate scope, giving examples and highlighting probable mechanistic pathways. Unfortunately, the manuscript is lacking somewhat in the introductory content; the background and prior art is not discussed in any great detail and the topic of the review is not surmised particularly well. There are also numerous errors in the schemes as well as the text. The work discussed within the review itself, albeit interesting, is also very niche. There are only a few examples presented and only 11 papers cited, the vast majority of which are from the authors own group. Hence, I do not find this work of sufficient value for publication in Royal Society Open Science.

Answer:

1. Heterocycles are ubiquitous in natural products, pharmaceuticals, organic materials, and numerous functional molecules. Therefore, organic chemists have been making extensive efforts to produce these heterocyclic compounds by developing new and efficient synthetic transformations. Recently, transition-metal-catalyzed reactions are one of the attractive methodologies for synthesizing heterocycles, however, transition-metal-catalyzed reactions are still limited in applications and confront challenges to some extent, since transition-metals are expensive, toxic, inconvenient for operation, and environment damage. In this regard, a transition metal-free methodology for the construction of an important heterocyclic compounds in drug discovery and material science has attracted attention. ROTf (alkyl trifluoromethanesulfonates) are powerful alkylating reagents for diverse substrates containing heteroatoms. The resulting triflates could be further transformed to other products by

substitution/addition/cyclization. In this review, we mainly classified the typical ROTf-induced annulation of N-reagents, S-reagents, O-reagents and P-reagents with themselves or alkynes to afford cyclic compounds as well as summarized their recent advances from different aspects. As suggested by Reviewers, we have made appropriate changes to the text and the mechanism in all schemes as shown in revision.

- 2. As questioned by Reviewer 3, we have made correction for Schemes and the text.**
- 3. We carefully checked the literature and more references were added appropriately in the revised manuscript.**

Appendix C

R. Soc. open sci. article template

ROYAL SOCIETY
OPEN SCIENCE

R. Soc. open sci.
doi:10.1098/not yet assigned

ROTf-induced annulation of heteroatom reagents and unsaturated substrates leading to cyclic compounds

Song Zou^{a#}, Sheng Wang^{a#}, Chanjuan Xi^{a,b*}

^aMOE Key Laboratory of Bioorganic Phosphorus Chemistry & Chemical Biology, Department of Chemistry, Tsinghua University, Beijing 100084, China

^bState Key Laboratory of Elemento-Organic Chemistry, Nankai University, Tianjin 300071, China

Keywords: trifluoromethanesulfonates; heteroatom reagents; annulation; cyclic compounds

1. Summary

The development of metal-free organic reactions is one of the hotspots in the synthesis of cyclic compounds. ROTf (alkyl trifluoromethanesulfonates), due to their good electrophilicity are powerful alkylating reagents at heteroatom such as nitrogen, oxygen, sulfur, and phosphorus to induce an electrophilic center for carbon-carbon or carbon-heteroatom bond formation. Inspired by this chemistry, a variety of researches concentrating on heterocycles synthesis has been carried out. In this review, we mainly summarize the ROTf-induced annulation of heteroatom reagents such as nitriles, carbodiimides, azobenzenes, isothiocyanates, aldehydes, isocyanates, and phosphalkene with themselves or alkynes to afford cyclic compounds.

2. Introduction

Heterocycles are ubiquitous in natural products, pharmaceuticals, organic materials, and numerous functional molecules. Therefore, organic chemists have been making extensive efforts to produce these heterocyclic compounds by developing new and efficient synthetic transformations. Transition-metal-catalyzed reactions are some of the most attractive methodologies for synthesizing heterocycles, since a transition-metal-catalyzed reaction can directly construct complicated molecules from readily accessible starting materials under mild conditions.[1] Nevertheless, transition-metal-catalyzed reactions are still limited in applications and confront challenges to some extent, since transition-metals are expensive, toxic, inconvenient for operation, and environment damage. In this regard, a transition metal-free methodology for the construction of an important heterocyclic compounds in drug discovery and material science has attracted attention [2].

ROTf (alkyl trifluoromethanesulfonates) are powerful alkylating reagents, which are frequently utilized in alkylation of nucleophiles [3]. Benefit from fluorine in ^-OTf to stabilize the negative charge, ^-OTf possess excellent leaving ability, which make the ROTf much more reactive as alkylation reagents than alkyl iodide and $(MeO)_2SO_2$. As useful and versatile precursors in a variety of organic transformations, many methods have been developed to prepare ROTf, such as reaction of orthoformates and triflic anhydrides under solvent-free condition [4].

The intrinsic electrophilicity of ROTf is often used as alkylation reagents for diverse substrates containing heteroatoms such as P [5], O [6], N [7-9], S [10,11], Te [12], Ge [13], Bi [14], Si [15], and Se [16,17], which straightforwardly afforded the corresponding alkylated products with aid of base or stable triflates irreversibly that even could be applied as ionic liquids [18,19]. Moreover, the resulting triflates could be further transformed to other products by substitution [20] (Scheme 1, a). On the other hand, unsaturated heteroatom-containing reagents, such as nitriles, aldehydes, isocyanates, and isothiocyanates, could be captured by ROTf to generate electrophiles bearing carbon cations [7,9,21], which could be subsequently reacted with appropriate unsaturated substrates to produce cyclic compounds by tandem electrophilic reactions/cyclization (Scheme 1, b). ROTf-induced cyclization featuring metal-free, easy to handle, and good selectivity provided a feasible approach to diverse cyclic compounds. In this review, we mainly focus on the ROTf-induced annulation of unsaturated heteroatom reagents such as nitriles, carbodiimides, azobenzenes, isothiocyanates, aldehydes, isocyanates, and phosphalkene with themselves or alkynes to afford cyclic compounds.

a. ROTf-mediated substitution:

b. ROTf-mediated cyclization:

Scheme 1. ROTf-mediated reactions.

3. ROTf-induced annulation of nitrogen-containing substrates with unsaturated compounds

3.1 ROTf-induced annulation of nitriles

Nitriles as unsaturated heteroatom reagents could react with ROTf to form *N*-alkylated nitriliums, which were well investigated by Booth [9] in 1980. However, electrophilicity of *N*-alkylated nitriliums has rarely been utilized in further reactions until recently. In 2014, our group [22] reported MeOTf-induced carboannulation of aryl nitriles and aromatic alkynes to construct indenones **1** (Scheme 2). Triflate **3** was isolated when 5-decyne was used indicating **I-1** might be an intermediate in this reaction. A range of functionalized indenone derivatives was obtained. When *ortho*-substituted aryl nitriles were utilized, indenone imine **I-2** would further cyclize with another molecule of nitriliums to give indeno[1,2-*c*]-isoquinolines **2** with construction of one carbocycle and

one heterocycle. Although transition-metal-catalyzed annulation of benzimide or arylcarbonyl and aryl nitrile with alkynes to formation of indenones has been reported [23], this reaction reveals a simple reaction process for the synthesis of indenones under metal-free.

Scheme 2. MeOTf-induced cyclization of aryl nitriles and alkynes.

Arylnitriles and alkynes could be induced by MeOTf to afford indenones *via* intermediate **I-5**. We envisioned the utilization of alkylnitriles, which lack an aryl group for the ring closure, might lead to a different *way* for ring formation. To our delight, reaction of alkylnitriles, alkynes, and MeOTf indeed afforded tetrasubstituted NH-pyrroles **4** *with* high regioselectivity. The structure includes one carbon from ROTf to join the pyrroles [24] (Scheme 3). It is noteworthy that the cyclized pyrrole captures another nitrilium leading to substituted 2-acyl-NH-pyrroles after hydrolysis. When EtOTf was utilized instead of MeOTf, the product **5** was obtained. *This reaction provides a practical and convenient method for the synthesis of multiply substituted 2-acylpyrroles from readily available starting materials in a one-pot reaction.*

Scheme 3. ROTf-induced cyclization of alkylnitriles and alkynes.

Furthermore, when 1,2-diphenylethyne and 1,2-di-*p*-tolylethyne were employed to react with MeOTf and alkylnitriles at 130°C , the isoquinolines **6** were obtained in good yields via intermediate **I-9**. The representative results are summarized in Scheme 4. In the cases, the Friede-Crafts reaction is favour to give 6-membered products

Scheme 4. MeOTf-induced cyclization of alkylnitriles and diarylalkynes

3.2 ROTf-induced annulation of carbodiimides

More recently, ROTf-induced electrophilic cyclization was extended to carbodiimides. Our group demonstrated an efficient ROTf-triggered intermolecular cyclization of carbodiimides to afford a range of 2-amino-4-imino-quinazolines **7**, 2-aminoquinazolinones **8**, and 2,4-diaminoquinazolines **9**, respectively, which are important motifs in pharmaceuticals [25]. When *N,N'*-diarylcarbodiimides were employed, the reaction proceeded smoothly to afford the corresponding 2-amino-4-imino-quinazolines **7**, which could be hydrolyzed to generate the

corresponding 2-aminoquinazolinones **8**. The representative results are shown in Scheme 5.

Scheme 5. $R'OTf$ -induced cyclization of diarylcarbodiimides

To further extend the substrate scope, a combination of two different carbodiimides has been achieved. A range of diarylcarbodiimides and dialkylcarbodiimides was investigated under the optimized reaction condition. The representative results are shown in Scheme 6. A plausible mechanism was proposed. First, the carbodiimide is methylated by MeOTf and subsequently attacked by another molecule of carbodiimide to give intermediate **I-11**. Then,

intramolecular nucleophilic attack takes place to afford the four-membered intermediate **I-12**, which generates carbenium **I-13** after ring opening via C-N bond cleavage. Finally, intramolecular Friedel-Crafts annulation occurs to form the corresponding quinazolinone imine **7**, which could give 2,4-diamino-quinazoline **9** (R = alkyl group) and 2-amino-quinazolinone **8** (R = aryl group) after hydrolysis. This annulation reaction appears a general entry to the synthesis of 2-amino-quinazolinones and 2,4-diamino-quinazolines in one-pot under metal-free.

Scheme 6. ROTf-induced cyclization of diarylcarbodiimides and dialkylcarbodiimides.

3.3 ROTf-induced annulation of azobenzenes

Apart from nitriles and carbodiimides, azobenzenes are also significant nitrogen-containing compounds, which possess Lewis **alkalinity**. In 2014, we demonstrated MeOTf-induced cyclization of azobenzenes by N=N bond cleavage with aid of TCQ (tetrachloro-**1,4**-benzoquinone) as oxidant to afford *N*-arylbenzimidazoles **10** [26] (Scheme 7). When unsymmetrical azobenzenes were used, cyclization tends to occur **on the** electron-rich anisoyl ring (**10e-10g**). EtOTf could facilitate N=N bond cleavage as well to generate 2-methylbenzimidazole **10h**. **Although the reaction mechanism is not clear,** a plausible mechanism **is** shown in Scheme 7. **The carbon atom from MeOTf inserts into the N=N bond and then cyclization to form *N*-arylbenzimidazole. This is the first example of N=N bond cleavage by a light main group element.**

Scheme 7. ROTf-induced cyclization of azobenzenes.

4. ROTf-induced annulation of sulfur-containing substrates with alkynes

4.1 ROTf-induced annulation of arylisothiocyanates

The isothiocyanates possess the chemical group $-N=C=S$, which represents versatile reactivity in the synthesis of nitrogen- or sulfur-containing heterocycles. Comparatively speaking, the sulfur atom is strongly nucleophilic. Recently, we reported MeOTf as an electrophile [27] to react with arylisothiocyanates and alkynes leading to diverse highly substituted quinolones. The representative results are shown in Scheme 8. A tandem electrophilic activation/cyclization via intermediate **I-18** is believed to be a possible process. This reaction demonstrated superiority on substrate scope for isothiocyanates. Furthermore, unsymmetrical alkynes such as terminal alkynes, (bromoethynyl) benzenes, even alkynes containing ester group could be applied in this reaction. In addition, alkyltriflates bearing C-C triple bond gave polycyclic quinolines **11d** via sequent cyclization process. Great effort has been paid in transformation of thioalkoxyl group such as oxidation, reduction, and cross-coupling reaction, which make this method more powerful in organic synthesis [27]. This reaction represents a concise, metal-free, and one-pot method for synthesis of functionalized quinolines.

Scheme 8. ROTf-induced cyclization of arylisothiocyanates and arylalkynes.

4.2 ROTf-induced annulation of alkyisothiocyanates

Substitution of arylisothiocyanates with alkyisothiocyanates that lack an aryl group for the ring closure, might lead to a new reaction mode. To our delight, the reaction proceeded well to afford indenone **12** after hydrolysis [28]. A range of arylalkynes could be employed in this reaction. The representative results are summarized in Scheme 9. A plausible mechanism is also described in Scheme 9. MeOTf as an electrophile reacts with isothiocyanate to form methylthio-substituted carbenium ion **I-19**, which followed the reaction with arylalkyne to form intermediate **I-20**. Utilization of the arylisothiocyanate affords

quinoline **11**. Without aryl group in the alkylisothiocyanate, the nucleophilicity-strong sulfur atom attacks carbenium of **I-20** to form four-membered thiete **I-21**, which could be followed by ring opening with the C-S bond cleavage to form carbenium **I-23** *via* intermediate **I-22**. Finally, intramolecular Friedel-Crafts reaction of **I-23** affords indenone imine **I-24**, which undergoes hydrolysis to form indenone **12**. This reaction represents the first example of cleavage C-S bond in the isothiocyanate for construction of the carbocyclic compound under metal-free.

Scheme 9. ROTf-induced annulation of R-N=C=S and arylalkynes.

More recently, Li and co-workers [29] reported MeOTf-induced intramolecular cyclization of isothiocyanates to afford 1-(methylthio)-3,4-dihydroisoquinolines **13** (Scheme 10). The reaction may process by a tandem electrophilic activation and intramolecular Friedel-Crafts reaction.

Scheme 10. MeOTf-induced intramolecular cyclization of isothiocyanates.

5. ROTf-induced annulation of oxygen-containing substrates

5.1 ROTf-induced annulation of aldehydes

Apart from S- and N-reagents that could react with ROTf straightforwardly, O-reagents also demonstrated good affinity with ROTf. As a part of ongoing projects on the alkyltriflate-triggered annulation, a reaction of MeOTf, aldehydes, and arylalkynes was investigated [30] and a variety of 2,3-disubstituted 1-indanones was obtained. The representative results are shown in Scheme 11. It is noteworthy that a catalytic amount of MeOTf was employed and the reaction proceeded in satisfactory yield. Although the reaction

mechanism is not clean, a plausible mechanism is shown in the bottom of Scheme 11. First, MeOTf as an electrophile reacts with aldehyde to afford the oxonium **I-25**, which couples with alkyne to form the highly active oxetanium intermediate **I-26** via [2+2] cycloaddition. Then, the intermediate **I-26** undergoes spontaneous isomerization to form the 4π -Nazarov intermediate **I-27**, followed by Nazarov cyclization to give 1-indanone **14** and regeneration of MeOTf. This reaction provides a practical and convenient method for the synthesis of 2,3-disubstituted 1-indanones from readily available starting materials via MeOTf-induced catalysis.

Scheme 11. MeOTf-induced cyclization of aldehyde with arylalkynes.

5.2 ROTf-induced annulation of arylisocyanates

Isocyanate is the functional group with the formula $R-N=C=O$, in which N and O may both be alkylated by ROTf. We investigated a

reaction of MeOTf, arylisocyanates, and arylalkynes [31]. Notably, a range of 4-methoxyl-2,3-diarylquinolines **15** was obtained in good yields and the representative results are shown in Scheme 12. Based on the results, a tandem [2+2] cycloaddition and intramolecular Friedel-Crafts reaction may be included in the reaction pathway. It is noteworthy that this reaction has limitations with only diarylalkynes and MeOTf for the construction of 4-methoxyl-2,3-diarylquinolines **15**.

Scheme 12. MeOTf-induced cyclization of arylisocyanates and arylalkynes.

Phenanthridinones are extensively found in natural products and bioactive molecules. We envisioned that MeOTf-induced intramolecular annulation of 2-phenyl aryllisocyanates would provide a pathway for the synthesis of phenanthridinones. During the course of our study on the CO₂ chemistry [32], we found a one-pot method for the synthesis of phenanthridinones [33] based on the MeOTf- and TBD-mediated carbonylation of *ortho*-arylanilines with CO₂. The representative results and reaction pathway are shown in Scheme 13. This reaction shows MeOTf-induced carbonylation reaction of *o*-arylanilines applying CO₂ as the ideal carbonyl source to synthesize phenanthridinones containing a free (NH)-lactam motif under metal-free conditions.

Scheme 13. MeOTf-induced cyclization with *o*-arylanilines and CO₂.

6. ROTf-induced annulation of phosphorus-containing substrates

In 2006, Gates and co-workers [34] utilized the strong electrophilicity of MeOTf in the synthesis of highly strained four-membered phosphorus heterocycles with phosphalkenes as precursor (Scheme 14). The structure of the unprecedented diphosphetanium salt **17** was identified by single crystal X-ray diffraction. Although the reaction mechanism is not clean, this reaction demonstrated a convenient method for synthesis of highly strained phosphorus

heterocycles, which may be used as a propagating species in the cationic polymerization of phosphalkenes.

Scheme 14. MeOTf-induced cyclization with phosphalkenes.

7. Conclusion and future outlook

ROTf has been a powerful reagent in organic synthesis featuring efficient, metal-free, and **easy of handling**. A range of heteroatom-containing unsaturated reagents such as nitriles, carbodiimides, azobenzenes, isothiocyanates, aldehydes, isocyanates, and phosphalkene could be alkylated by the ROTf to generate reactive intermediates, which are capable of capturing other **electrophilic** substrates to afford cyclic compounds with a rational design. The synthetic methodology to various carbocyclic and heterocyclic compounds **has been** widely developed. Although ROTf has exhibited great value for the synthesis of cyclic compounds, further exploration is required for the utilization of functionalized ROTf reagents and other heteroatom reagents such as organoselenium, organophosphorus **substrates for synthesis of functionalized compounds**. Furthermore, four-member ring intermediates proposed to elucidate surprising rearrangements **still needs** firm evidence. We anticipate that ROTf can be extended to more organic reactions in organic synthesis.

Data accessibility

As a review article, there is no original data included.

Acknowledgements

We thank all the cited authors for providing the research results for our reference.

Funding

This work was supported by the National Natural Science Foundation of China (nos. 21472106 and 91645120).

Competing Interests

We declare we have no competing interests.

Authors' Contributions

S. Zou and S. Wang participated in the design of the study and drafted the manuscript; [#]S. Zou and S. Wang contributed equally; C. Xi conceived the study, coordinated the study and helped draft the manuscript. All authors gave final approval for publication.

References

1. (a) Lie JJ, Gribble GW. Palladium in Heterocyclic Chemistry, Pergamon Press, New York, 2000. (ISBN: 9780080451169); (b) Zeni G, Larock RC. 2004 Synthesis of Heterocycles via Palladium π -olefin and π -alkyne chemistry. *Chem. Rev.* **104**, 2285-2309. (doi:10.1021/cr020085h); (c) Nakamura I, Yamamoto Y. 2004 Transition-metal-catalyzed reactions in heterocyclic synthesis. *Chem. Rev.* **104**, 2127-2198. (doi:10.1021/cr020095i); (d) Zeni G, Larock RC. 2006 Synthesis of heterocycles via palladium-catalyzed oxidative addition. *Chem. Rev.* **106**, 4644-4680. (doi:10.1021/cr0683966); (e) Álvarez-Corral M, Muñoz-Dorado M, Rodríguez-García I. 2008 Silver-mediated synthesis of heterocycles. *Chem. Rev.* **108**, 3174-3198. (doi:10.1021/cr078361l); (f) Patil NT, Yamamoto Y. 2008 Coinage metal-assisted synthesis of heterocycles. *Chem. Rev.* **108**, 3395-3442. (doi:10.1021/cr050041j).
2. (a) Li CJ, Trost BM. 2008 Green chemistry for chemical synthesis. *Proc. Natl. Acad. Sci. USA* **105**, 13197-13202. (doi:10.1073/pnas.0804348105); (b) Antonchick AP, Samanta R, Kulikov K, Lategahn J. 2011 Organocatalytic, oxidative, intramolecular C-H bond amination and metal-free cross-amination of unactivated arenes at ambient temperature. *Angew. Chem, Int. Ed.* **50**, 8605-8608. (doi:10.1002/anie.201102984); (c) Kim HJ, Kim J, Cho SH, Chang S. 2011 Intermolecular oxidative C-N bond

formation under metal-free conditions: control of chemoselectivity between aryl sp^2 and benzylic sp^3 C-H bond imidation. *J. Am. Chem. Soc.* **133**, 16382-16385. (doi:10.1021/ja207296y); (d) Kantak AA, Potavathri S, Barham RA, Romano KM, DeBoef B. 2011 Metal-free intermolecular oxidative C-N bond formation via tandem C-H and N-H bond functionalization. *J. Am. Chem. Soc.* **133**, 19960-19965. (doi:10.1021/ja2087085); (e) Froehr T, Sindlinger CP, Kloeckner U, Finkbeiner P, Nachtsheim BJ. 2011 A metal-free amination of benzoxazoles-the first example of an iodide-catalyzed oxidative amination of heteroarenes. *Org. Lett.* **13**, 3754-3757. (doi:10.1021/ol201439t); (f) Xiao Q, Tian L, Tan R, Xia Y, Qiu D, Zhang Y, Wang J. 2012 Transition-metal-free electrophilic amination of arylboroxines. *Org. Lett.* **14**, 4230-4233. (doi:10.1021/ol301912a); (g) Sun. CL, Shi. ZJ. 2014 Transition-metal-free coupling reactions. *Chem. Rev.* **114**, 9219-9280. (doi:10.1021/cr400274j); (h) Kim YR, Cho S, Lee PH. 2014 Metal-free azaphosphaannulation of phosphonamides through intramolecular oxidative C-N bond formation. *Org. Lett.* **16**, 3098-3101. (doi:10.1021/ol501207w); (i) Youn SW, Lee EM. 2016 Metal-free one-pot synthesis of N,N' -diarylamidines and N -arylbenzimidazoles from arenediazonium salts, nitriles, and free Anilines. *Org. Lett.* **18**, 5728-5731. (doi:10.1021/acs.orglett.6b02994).

3. (a) Alder RW, Phillips JGE, Huang L, Huang X. 2005

- “Methyltrifluoromethanesulfonate”, *encyclopedia of reagents for organic synthesis*, John Wiley & Sons. (doi:10.1002/047084289X.rm266m.pub2); (b) Stang PJ, Hanack M, Subramanian LR. 1982 Perfluoroalkanesulfonic esters: methods of preparation and applications in organic chemistry. *Synthesis*. **2**, 0085-0127. (doi: 10.1055/s-1982-29711); (c) Howells RD, McCown JD. 1977 Trifluoromethanesulfonic acid and derivatives. *Chem. Rev.* **77**, 69-92. (doi:10.1021/cr60305a005)
4. (a) Zheng D, Ma H, Ding K. 2017 A practical synthesis of trifluoromethanesulfonate esters. *Chin. J. Org. Chem.* **37**, 1582-1584. (doi: 10.6023/cjoc201701035); (b) Chapman RD, Andreshak JL, Herrlinger SP, Shackelford SA, Hildreth RA, Smith JP. 1986 Trifluoromethanesulfonate esters from dibromoalkane metatheses with silver triflate: mechanistic and synthetic aspects. *J. Org. Chem.* **51**, 3792-3798. (doi:10.1021/jo00370a009); (c) Baraznenok IL, Nenajdenko VG, Balenkova ES. 2000 Chemical transformations induced by triflic anhydride. *Tetrahedron*. **56**, 3077-3119. (doi:10.1016/S0040-4020(00)00093-4); (d) Aubert C, Begue JP. 1985 A simple preparation of alkyl trifluoromethanesulfonates (triflates) from alkyl trimethylsilyl ethers. *Synthesis*. **8**, 759-760. (doi:10.1055/s-1985-31336); (e) Burdon J, McLoughlin VCR. 1965 Trifluoromethanesulphonate esters and their alkylating properties. *Tetrahedron*. **21**, 1-4. (doi:10.1016/S0040-4020(01)82194-3); (f) Dobbs AP, Jones K, Veal KT. 1997 The generation and cyclisation of pyridinium radicals as a potential

- route to indolizidine alkaloids. *Tetrahedron Lett.* **38**, 5383-5386. (doi: 10.1016/S0040-4039(97)01178-7); (g) Beard CD, Baum K, Grakauskas V. 1973 Synthesis of some novel trifluoromethanesulfonates and their reactions with alcohols. *J. Org. Chem.* **38**, 3673-3677. (doi:10.1021/jo00961a003); (h) Ignatyev NV, Barthen P, Kucheryna A, Willner H, Sartori P. 2012 A method for producing perfluoroalkanesulfonic acid esters. *Molecules.* **17**, 5319-5337. (doi:10.3390/molecules17055319)
5. Burford N, Dyker CA, Decken A. 2005 Facile synthetic methods for the diversification of catena-polyphosphorus cations. *Angew. Chem. Int. Ed.* **44**, 2364-2367. (doi:10.1002/anie.200462997)
 6. Imamoto T, Kikuchi SI, Miura T, Wada Y. 2001 Stereospecific reduction of phosphine oxides to phosphines by the use of a methylation reagent and lithium aluminum hydride. *Org. Lett.* **3**, 87-90. (doi: 10.1021/ol0068041)
 7. Rong MK, Duin K, Dijk T, Pater JJM, Deelman B.-J, Nieger M, Ehlers AW, Slootweg JC, Lammertsma K. 2017 Iminophosphanes: synthesis, rhodium complexes, and ruthenium(II)-catalyzed hydration of nitriles. *Organometallics.* **36**, 1079-1090. (doi:10.1021/acs.organomet.7b00057)
 8. Ye H, Liu R, Li D, Liu Y, Yuan H, Guo W, Zhou L, Cao X, Tian H, Shen J, Wang P. 2013 A safe and facile route to imidazole-1-sulfonyl azide as a diazotransfer reagent. *Org. Lett.* **15**, 18-21. (doi:10.1021/ol3028708)

9. Booth BL, Jibodu KO, Proenca MF. 1980 The synthesis and some reactions of *N*-methylnitrium trifluoromethanesulphonate salts. *J. Chem. Soc., Chem. Commun.* **0**, 1151-1153. (doi:10.1039/C39800001151)
10. Thomas Z, Gregor L. 1998 Synthesis of β -mannosides via prearranged glycosides. *Angew. Chem. Int. Ed.* **37**, 3129-3132. (doi:10.1002/(SICI)1521-3773(19981204)37:22<3129::AID-ANIE3129>3.0.CO;2-8)
11. Ranade SC, Kaeothip S, Demchenko AV. 2010 Glycosyl alkoxythioimidates as complementary building blocks for chemical glycosylation. *Org. Lett.* **12**, 5628-5631. (doi:10.1021/ol1023079)
12. Laali K, Chen HY, Gerzina RJ. 1988 Methylation of aromatics with Me_3S^+ , Me_3Se^+ , and Me_3Te^+ in superacid media. *J. Org. Chem.* **348**, 199-204. (doi:10.1016/0022-328X(88)80396-6)
13. Su B, Ganguly R, Li Y, Kinjo R. 2016 Synthesis, characterization, and electronic structures of a methyl germyliumylidene ion and germylone-group VI metal complexes. *Chem. Commun.* **52**, 613-616. (doi:10.1039/C5CC08665E)
14. Wallenhauer S, Seppelt K. 1994 Methylbismuth(V) compounds. *Angew. Chem. Int. Ed.* **33**, 976-978. (doi:10.1002/anie.199409761)
15. Yamaguchi T, Asay M, Sekiguchi A. 2012 $[[(\text{Me}_3\text{Si})_2\text{CH}]_2^i\text{PrSi}(\text{NHC})\text{Si}=\text{Si}(\text{Me})\text{Si}^i\text{Pr}[\text{CH}(\text{SiMe}_3)_2]_2]^+$: A molecule with disilyl cation character. *J. Am. Chem. Soc.* **134**, 886-889. (doi:10.1021/ja210669n)
16. Spera ML, Harman WD. 1999 Osmium-mediated electrophilic

- addition reactions with selenophene and activation of the Se-C bond. *Organometallics*. **18**, 1559-1561. (doi:10.1021/om9806949)
17. Mutoh Y, Murai T. 2003 Acyclic selenoiminium salts: isolation, first structural characterization, and reactions. *Org. Lett.* **5**, 1361-1364. (doi:10.1021/ol034334f)
18. Chen ZJ, Xi HW, Lim KH, Lee JM. 2013 Distillable ionic liquids: reversible amide O alkylation. *Angew. Chem. Int. Ed.* **52**, 13392-13396. (doi:10.1002/anie.201306476)
19. Li G, Xue Z, Cao B, Yan C, Mu T. 2016 Preparation and properties of C=X (X: O, N, S) based distillable ionic liquids and their application for rare earth separation. *ACS. Sustain. Chem. Eng.* **4**, 6258-6262. (doi:10.1021/acssuschemeng.6b01984)
20. (a) Ye H, Liu R, Li D, Liu Y, Yuan H, Guo W, Zhou L, Cao X, Tian H, Jie Shen J, Wang PG. 2013 A safe and facile route to imidazole-1-sulfonyl azide as a diazotransfer reagent. *Org. Lett.* **15**, 18-21. (doi: 10.1021/ol3028708); (b) Ulibarri G, Choret N, Bigg DCH. 1996 Activation of imidazolides using methyl trifluoromethanesulfonate: a convenient method for the preparation of hindered esters and amides. *Synthesis*. **11**, 1286-1288. (doi:10.1055/s-1996-4399); (c) Zhu F, Tao J, Wang Z. 2015 Palladium-catalyzed C-H arylation of (benzo)oxazoles or (benzo)thiazoles with aryltrimethylammonium triflates. *Org. Lett.* **17**, 4926-4929. (doi:10.1021/acs.orglett.5b02458); (d) Yu W, Yang S, Xiong F, Fan T, Feng Y, Huang Y, Fu J, Wang T. 2018

Palladium-catalyzed carbonylation of benzylic ammonium salts to amides and esters via C–N bond activation. *Org. Biomol. Chem.* **16**, 3099-3103. (doi:10.1039/C8OB00488A)

21. (a) Dondoni A, Catozzi N, Marra A. 2005 Concise and practical synthesis of *c*-glycosyl ketones from sugar benzothiazoles and their transformation into chiral tertiary alcohols. *J. Org. Chem.* **70**, 9257-9268. (doi:10.1021/jo051377w); (b) Murai T, Mutoh Y, Ohta Y, Murakami M. 2004 Synthesis of tertiary propargylamines by sequential reactions of in situ generated thioiminium salts with organolithium and -magnesium reagents. *J. Am. Chem. Soc.* **126**, 5968-5969. (doi:10.1021/ja048627v)
22. Yan X, Zou S, Zhao P, Xi C. 2014 MeOTf-induced carboannulation of aryl nitriles and aromatic alkynes: a new metal-free strategy to construct indenones. *Chem. Commun.* **50**, 2775-2777. (doi:10.1039/C4CC00088A)
23. For examples, see: (a) Pletnev AA, Tian Q, Larock RC. 2002 Carbopalladation of nitriles: synthesis of 2,3-diaryllindenones and polycyclic aromatic ketones by the Pd-catalyzed annulation of alkynes and bicyclic alkenes by 2-iodoarenenitriles *J. Org. Chem.*, **67**, 9276-9287. (doi:10.1021/jo026178g); (b) Miura T, Murakami M. 2005 Rhodium-catalyzed annulation reactions of 2-cyanophenylboronic acid with alkynes and strained alkenes. *Org. Lett.*, **7**, 3339-3341. (doi:10.1021/ol051249u); (c) Tsukamoto H, Kondo Y. 2007 Palladium(II)-catalyzed annulation of alkynes with *ortho*-ester-containing phenylboronic acids. *Org. Lett.* **9**, 4227-

4230. (doi:10.1021/ol701776m); (d) Harada Y, Nakanishi J, Fujihara H, Tobisu M, Fukumoto Y, Chatani N. 2007 Rh(I)-catalyzed carbonylative cyclization reactions of alkynes with 2-bromophenylboronic acids leading to indenones. *J. Am. Chem. Soc.* **129**, 5766-5771. (doi:10.1021/ja070107n); (e) Morimoto T, Yamasaki K, Hirano A, Tsutsumi K, Kagawa N, Kakiuchi K, Harada Y, Fukumoto Y, Chatani N, Nishioka T. 2009 Rh(I)-catalyzed CO gas-free carbonylative cyclization reactions of alkynes with 2-bromophenylboronic acids using formaldehyde. *Org. Lett.* **11**, 1777-1780. (doi:10.1021/ol900327x); (f) Li BJ, Wang HY, Zhu QL, Shi ZJ, 2012 Rhodium/copper-catalyzed annulation of benzimides with internal alkynes: indenone synthesis through sequential C-H and C-N cleavage *Angew. Chem. Int. Ed.* **51**, 3948-3952. (doi:10.1002/anie.201200271); (g) Chen S, Yu J, Jiang Y, Chen F, Cheng J. 2013 Rhodium-catalyzed direct annulation of aldehydes with alkynes leading to indenones: proceeding through *in situ* directing group formation and removal. *Org. Lett.* **15**, 4754-4757. (doi:10.1021/ol4021145); (h) Qi Z, Wang M, Li X. 2013 Access to Indenones by rhodium(III)-catalyzed C-H annulation of aryl nitrones with internal alkynes. *Org. Lett.* **15**, 5440-5443. (doi: 10.1021/ol4025309).

24. Liu Y, Yi X, Luo X, Xi C. 2017 MeOTf-mediated annulation of alkylnitriles and arylalkynes leading to polysubstituted NH-pyrroles. *J. Org. Chem.* **82**, 11391-11398.

(doi:10.1021/acs.joc.7b01845)

25. Zhang X, Wang S, Liu Y, Xi C. 2018 Triflates-triggered intermolecular cyclization of carbodiimides leading to 2-aminoquinazolinone and 2,4-diaminoquinazoline derivatives. *Org. Lett.* **20**, 2148-2151. (doi:10.1021/acs.orglett.8b00314)
26. Yan X, Yi X, Xi C. 2014 Direct cleavage of the N=N bond of azobenzenes by MeOTf leading to *N*-arylbenzimidazoles. *Org. Chem. Front.* **1**, 657-660. (doi:10.1039/C4QO00056K)
27. Zhao P, Yan X, Yin H, Xi C. 2014 Alkyltriflate-triggered annulation of arylisothiocyanates and alkynes leading to multiply substituted quinolines through domino electrophilic activation. *Org. Lett.* **16**, 1120-1123. (doi:10.1021/ol500221u)
28. Zhao P, Liu Y, Xi C. 2015 MeOTf-induced carboannulation of isothiocyanates and aryl alkynes with C=S bond cleavage: access to indenones. *Org. Lett.* **17**, 4388-4391. (doi:10.1021/acs.orglett.5b02201)
29. Wen L, Dou Q, Wang Y, Zhang J, Guo W, Li M. 2017 Synthesis of 1-thio-substituted isoquinoline derivatives by tandem cyclization of isothiocyanates. *J. Org. Chem.* **82**, 1428-1436. (doi:10.1021/acs.joc.6b02605)
30. Liu Y, Zhao P, Zhang B, Xi C. 2016 MeOTf-catalyzed annulation of aldehydes and arylalkynes leading to 2,3-disubstituted indanones. *Org. Chem. Front.* **3**, 1116-1119. (doi:10.1039/C6QO00253F)
31. Liu Y, Zhang X, Xi C. 2018 MeOTf-induced annulation of

- arylisocyanates and arylalkynes leading to 4-methoxy-2,3-diarylquinolines. *Tetrahedron Lett.* **25**, 2440-2442. (doi:10.1016/j.tetlet.2018.05.030)
32. (a) Shao P, Wang S, Chen C, Xi C. 2015 Zirconocene-catalyzed sequential ethylcarboxylation of alkenes using ethylmagnesium chloride and carbon dioxide. *Chem. Commun.* **51**, 6640-6642. (doi:10.1039/C5CC01153A); (b) Wang S, Shao P, Chen C, Xi C. 2015 Copper-catalyzed carboxylation of alkenylzirconocenes with carbon dioxide leading to α,β -unsaturated carboxylic acids. *Org. Lett.* **17**, 5112-5115. (doi:10.1021/acs.orglett.5b02619); (c) Wang S, Du G, Xi C. 2016 Copper-catalyzed carboxylation reactions using carbon dioxide. *Org. Biomol. Chem.* **14**, 3666-3676. (doi:10.1039/C6OB00199H); (d) Shao P, Wang S, Chen C, Xi C. 2016 Cp₂TiCl₂-catalyzed regioselective hydrocarboxylation of alkenes with CO₂. *Org. Lett.* **9**, 2050-2053. (doi:10.1021/acs.orglett.6b00665).
33. Wang S, Shao P, Du G, Xi C. 2016 MeOTf- and TBD-mediated carbonylation of ortho-arylanilines with CO₂ leading to phenanthridinones. *J. Org. Chem.* **81**, 6672-6676. (doi:10.1021/acs.joc.6b01318)
34. Bates JI, Gates PD, 2006 Diphosphiranium (P₂C) or diphosphetanium (P₂C₂) cyclic cations: different fates for the electrophile-initiated cyclodimerization of a phosphalkene. *J. Am. Chem. Soc.* **128**, 15998-15999. (doi:10.1021/ja0667662)

Appendix D

Respond to Editor and Referees

Respond to the Editor

Ethics statement: our work is not relevant to the point of ethics statement

Respond to the Reviewer 3

Comments to the Author(s)

This resubmitted manuscript is significantly improved from its previous form and the authors have addressed the majority of the issues raised by the reviewers on the initial submission. I do think that the document is somewhat lacking in the conclusions section, it would be better to see more of a comparative summary which highlights the benefits of these methodologies over the current state-of-the-art. Why is this approach better? What unique reactivities are there? Does the methodology provide access to unique architectures? Where will these methodologies lead in the future?

As this is an invited review I am inclined to accept this minireview with the caveat that the authors expand somewhat upon their closing remarks.

Why is this approach better?

Answer:

Heterocycles are ubiquitous in natural products, pharmaceuticals, organic materials, and numerous functional molecules. Therefore, organic chemists have been making extensive efforts to produce these heterocyclic compounds by developing new and efficient synthetic transformations. Recently, transition-metal-catalyzed reactions are one of the attractive methodologies for synthesizing heterocycles, however, transition-metal-catalyzed reactions are still limited in applications and confront challenges to some extent, since transition-metals are expensive, toxic, inconvenient for operation, and environment damage. In this regard, a transition metal-free methodology for the construction of an important heterocyclic compounds in drug discovery and material science has attracted attention. ROTf (alkyl trifluoromethanesulfonates) are powerful alkylating reagents for diverse substrates containing heteroatoms. The resulting triflates could be further transformed to other products by substitution/addition/cyclization. In addition , construction of heterocycles using ROTf is featuring efficient, metal-free, and easy of handing. These points are mentioned in the introduction and conclusion parts of revision

What unique reactivities are there?

Answer:

ROTf-induced cyclization featuring metal-free, easy to handle, and good selectivity provided a feasible approach to diverse cyclic compounds. These points are mentioned in the end of introduction part.

Does the methodology provide access to unique architectures?

Answer:

ROTf-induced annulation could provide various 5- and 6-membered heterocycles, most of them are important ubiquitous in natural products, pharmaceuticals, organic materials. For example, ROTf-triggered intermolecular cyclization of carbodiimides to provide 2-aminoquinazolinones and 2,4-diaminoquinazolines, which possess vital pharmacological activity. One more example is that MeOTf-induced direct cleavage of the N=N bond of azobenzenes leading to N-arylbenzimidazoles. In this reaction, the methyl carbon atom inserts into the N=N bond to form N-arylbenzimidazoles via cleavage of the N=N bond and two sp³ C–H bonds as well as one sp² C–H bond and meanwhile formation of three C–N bonds without the assistance of any metals or metalloids. The examples are mentioned in the text.

Where will these methodologies lead in the future?

Answer:

ROTf has been a powerful reagent in organic synthesis featuring efficient, metal-free, and easy of handling.

Although ROTf has exhibited great value for the synthesis of cyclic compounds, further exploration is required for the utilization of functionalized ROTf reagents and other heteroatom reagents such as organoselenium, organophosphorus substrates for synthesis of functionalized compounds.

Among the utilization of carboncations, ROTf induced unsaturated heteroatom-containing reagents to generate electrophiles bearing

carboncations, which could be involved new approaches in the synthesis of heterocycles under mild and metal-free conditions.

ROTF-induced [2+2] cycloaddition to afford four-member ring intermediates proposed to elucidate surprising rearrangements still needs firm evidence.

These points are mentioned in the conclusion and future outlook part.